# Local-scale Arctic tundra heterogeneity affects regional-scale carbon dynamics

M. J. Lara [1,2,3 ✉], A. D. McGuire[3], E. S. Euskirchen [3], H. Genet [3], S. Yi[4], R. Rutter[3], C. Iversen [5], V. Sloan[6] & S. D. Wullschleger[5]

In northern Alaska nearly 65% of the terrestrial surface is composed of polygonal ground, where geomorphic tundra landforms disproportionately influence carbon and nutrient cycling over fine spatial scales. Process-based biogeochemical models used for local to Pan-Arctic projections of ecological responses to climate change typically operate at coarse-scales (1km$^2$–0.5°) at which fine-scale (<1km$^2$) tundra heterogeneity is often aggregated to the dominant land cover unit. Here, we evaluate the importance of tundra heterogeneity for representing soil carbon dynamics at fine to coarse spatial scales. We leveraged the legacy of data collected near Utqiaġvik, Alaska between 1973 and 2016 for model initiation, parameterization, and validation. Simulation uncertainty increased with a reduced representation of tundra heterogeneity and coarsening of spatial scale. Hierarchical cluster analysis of an ensemble of 21$^{st}$-century simulations reveals that a minimum of two tundra landforms (dry and wet) and a maximum of 4km$^2$ spatial scale is necessary for minimizing uncertainties (<10%) in regional to Pan-Arctic modeling applications.

[1] Plant Biology Department, University of Illinois, Urbana, IL 61801, USA. [2] Geography Department, University of Illinois, Urbana, IL 61801, USA. [3] Institute of Arctic Biology, University of Alaska, Fairbanks, AK 99775, USA. [4] Institute of Fragile Ecosystem and Environment, School of Geographic Science, Nantong University, Nantong, China. [5] Environmental Sciences Division and Climate Change Science Institute, Oak Ridge National Laboratory, Oak Ridge, TN, USA. [6] School of Civil, Aerospace and Mechanical Engineering, Queens's Building, University of Bristol, Bristol, UK. ✉email: mjlara@illinois.edu

There are considerable uncertainties regarding the fate of permafrost carbon pools with projected warming over the next century. Warming[1], thawing[1,2], and degrading permafrost[2] have increased the vulnerability of modern and ancient soil carbon to decomposition[3]. Although evidence suggests that substantial losses of permafrost carbon may be inevitable[4], the pace of change spanning the Pan-Arctic is likely to vary regionally with climate change, vegetation composition, landform, soil carbon density, and ground ice content[5–8]. Therefore, uncertainties may be amplified in coarse-scale Pan-Arctic projections by the limited representation of local-scale processes, intrinsically linked with the mosaic of tundra landforms that profoundly influence ecosystem structure and function[9–11].

The tundra on the Arctic Coastal Plain of Alaska is highly heterogeneous (Fig. 1), nearly 65% of this landscape is composed of an intricate network of ice-wedge polygon landforms (Fig. 2)[12,13] developed by ground ice aggregation and degradation associated with frost heaving and ground subsidence[14]. The spatial distribution of these low relief (<0.5 m) landforms varies over small spatial scales (10–100 m²), strongly influencing surface and subsurface hydrology[2,15], snow distribution and depth[16–18], vegetation composition[19–21], soil carbon and nitrogen[22–25], carbon dioxide and methane fluxes[9,26–29], and an array of soil-forming processes[30]. Despite mounting evidence of the fine-scale microtopographic and landform-specific controls on ecosystem processes that govern ecosystem function, data limitations in the Arctic have restricted our ability to understand the importance of scale-, process-, and landform-dependent responses to global change.

Here we leveraged the legacy of measurements collected (described in "Model initialization") between 1973 to 2016 from a data-rich subregion of the Arctic Coastal Plain (i.e., Barrow Peninsula) to comprehensively parameterize the diverse mosaic of all dominant polygonal tundra landforms within a process-based terrestrial ecosystem model with dynamic organic soil layers (DOS-TEM). DOS-TEM is an intermediate-scale model capable of simulating carbon, nitrogen, and water cycles, with interacting permafrost dynamics[31]. DOS-TEM has been extensively validated and applied at a range of spatial scales[32–34]. Due to the strong control of polygonal landforms on ecosystem structure and function, we hypothesize that the local-scale representation of tundra heterogeneity (defined here as the total number of landforms represented in the model) will markedly influence regional-scale soil carbon projections. However, the potential impact of tundra heterogeneity and model spatial scale (i.e., grid size) on

Arctic tundra carbon dynamics remains highly uncertain, yet paramount for reducing model uncertainties spanning the Pan-Arctic. We evaluate the error of prediction in twenty-first century Arctic soil carbon stocks associated with tundra heterogeneity and spatial scale by running parallel DOS-TEM simulations with a range of resolved tundra landforms (6–1) and spatial scales (30 m–25 km²). Results indicate model error will be significantly constrained by representing a minimum of two tundra landforms (dry and wet) at a maximum model spatial scale of ≤4 km². However, all efforts to advance the representation of local-scale heterogeneity in terrestrial and earth system models will significantly improve global climate change projections in response to thawing and degrading permafrost carbon.

## Results

**Study region**. The Barrow Peninsula (~1800 km²) is situated on the northern limit of the Arctic Coastal Plain (Fig. 1). The mean annual temperature, precipitation, and snowfall are −11.2 °C, 115 mm, and 958 mm, respectively (1981–2010)[35] and the maximum thaw depth ranges from 30 to 90 cm[36,37]. This continuous permafrost region is characterized by meso-scale (tens to hundreds of square kilometers) drained thaw lake basins (DTLBs)

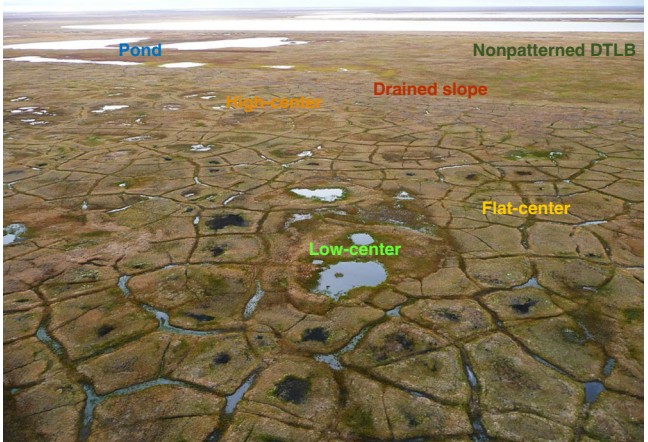

**Fig. 2 Oblique aerial photograph of the dominant polygonal tundra landforms on the Arctic Coastal Plain of Alaska.** Photograph acquired in August 2008 from southeast (~135° azimuth) of 71°16′46.02″N, 156° 25′45.35″.

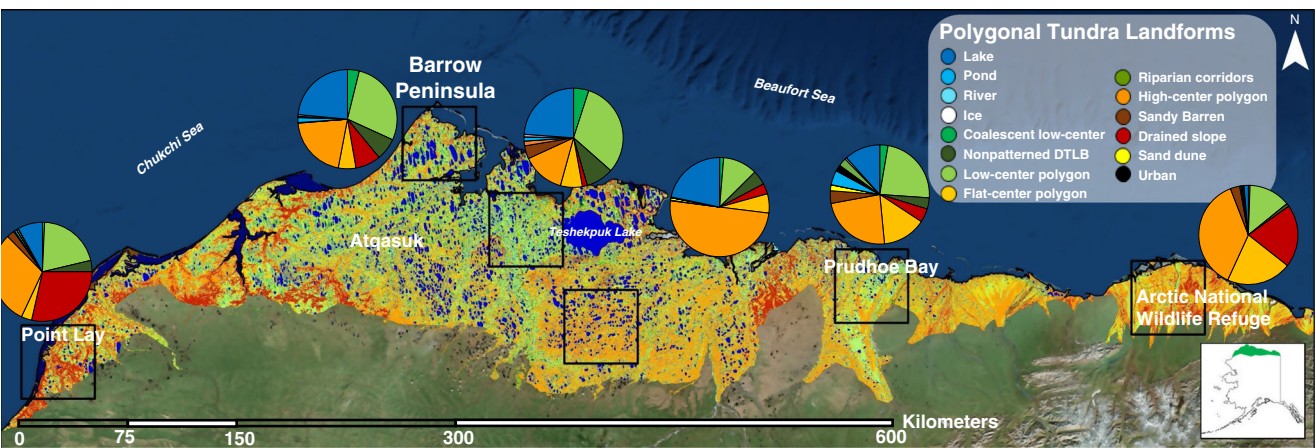

**Fig. 1 Heterogeneous distribution of polygonal tundra landforms on the Arctic Coastal Plain of Alaska.** The polygonal tundra map[10,13] is projected over ArcGIS World Imagery basemap (Sources: Esri, DigitalGlobe, GeoEye, i-cubed, USDA FSA, USGS, AEX, Getmapping, Aerogrid, IGN, IGP, swisstopo, and the GIS User Community).

and interstitial tundra[9,38], which are composed of a mosaic of fine-scale polygonal tundra landforms (tens to hundreds of square meters). Excluding lakes and rivers, the dominant polygonal tundra landforms in this region includes low-center (LC) polygon, flat-center (FC) polygon, high-center (HC) polygon, coalescent LC polygon, drained slopes (DS), nonpatterned DTLB (nDTLB), and thermokarst ponds, which cover an estimated 34, 24, 16, 11, 11, 3, and 1% of the land surface area, respectively[9,13]. Due to the similarity in morphological and physiological characteristics of coalescent LC polygons and thermokarst ponds, they are rarely differentiated in field observations. Therefore, both these landforms are combined and referred to as Ponds in the proceeding analysis. Though multiple vegetation communities may be found on each tundra landform, communities typically assemble along a soil moisture gradient representative of each landform[21]. These community–landform associations are identified as follows: dry Salix heath–DS, dry Luzula heath–HC, moist–wet Carex–Oncophorus meadow–FC, moist–wet Carex–Eriophorum meadow–LC, wet Dupontia meadow–nDTLB, and wet Arctophila pond margin–Pond[21].

**Model parameterization and validation**. We synthesized an extensive collection of field data measured on the Barrow Peninsula to parameterize and validate DOS-TEM (Supplementary Table 1 and Fig. 3). The majority of this data was acquired by scientific initiatives: (1) International Biological Research Program during the early 1970s[21,38–40], (2) Next Generation Ecosystem Experiments between 2010 and 2016[41–47], and (3) Carbon in Arctic Reservoirs Vulnerability Experiment (CARVE) during 2011–2015[48]. In addition, we leveraged key ancillary datasets including: soil carbon pedons (i.e., 100 cm soil cores)[22–24,30,38,39,49–51], vegetation carbon and nitrogen[21,38–40,52], eddy covariance measurements[48], and polygonal tundra landform maps[9,13].

Modeled carbon fluxes were compared to net ecosystem exchange (NEE) measurements from the CARVE tower near Utqiaġvik (71°19′22.72″N, 156°35′47.74″W). The tower footprint (~250 m radius) was located in a heterogeneous tundra site composed of all dominant polygonal tundra landforms (exception of Ponds). Although we identified good correspondence with modeled and measured NEE for most of our observations, DOS-TEM underestimated respiratory losses during the zero-curtain

seasonal freeze and thaw isothermal period (e.g., September and October)[53], resulting in an underestimate of the 1 to 1 line ($R^2 = 0.46$, $p < 0.001$, Fig. 3a, b). Simulated carbon pools for each tundra landform were compared and validated to (i) model benchmarks (Supplementary Table 1) and (ii) an independent subset of soil carbon pools (i.e., pedons), identifying excellent correspondence with modeled and measured carbon pools (Fig. 3c). Together these results demonstrate the ability of DOS-TEM to capture seasonal and inter-annual patterns of carbon dynamics in tundra ecosystems.

**Response of soil carbon to climate change**. We simulated the response of soil carbon pools to climate change within frozen and seasonally thawed organic (fibric and humic) and mineral horizons. These soil horizons are vertically stratified and vary in the degree of organic matter decomposition. Due to the known data limitations across the Arctic[54–56], the complete representation of tundra heterogeneity has not been possible. We used high (i.e., Canadian Centre for Climate Modeling and Analysis (CCCMA) A2) and low (i.e., ECHAM5 B1) climate and emission scenarios, comparable to the five best-performing CMIP5 (Coupled Model Intercomparison Project phase 5) model mean representative concentration pathways 6.0 and 4.5 for the Barrow Peninsula (Supplementary Figs. 1 and 2). Scenarios CCCMA A2 and ECHAM5 B1 project an increase in air temperature (6.96–5.72 °C, precipitation (182–215 mm), and atmospheric $CO_2$ (509–214 ppm) between 1970 and 2100.

Model simulations indicate that all landforms will gain soil carbon by the end of the twenty-first century (Fig. 4a). However, the trajectories between wetter versus dryer landforms diverged between climate change scenarios. Dependent on climate scenarios from 1970 to 2100, simulated soil C stocks in wet nDTLBs and LC increased between 6127 and 4012 (11.9–7.8%) and between 5246 and 3715 (8.5–6.0%) g C m$^{-2}$, respectively (Fig. 4a). Moist FC and aquatic Ponds gained soil carbon at a slightly lower rate than wet landforms, increasing between 2838 and 2073 (5.3–3.9%) and between 4762 and 3970 (7.2–6.0%) g C m$^{-2}$, respectively. While dry HC and DS landforms slowly increased in soil carbon content between 2389 and 1895 (3.6–2.8%) and between 2596 and 1834 (3.2–2.3%) g C m$^{-2}$, respectively.

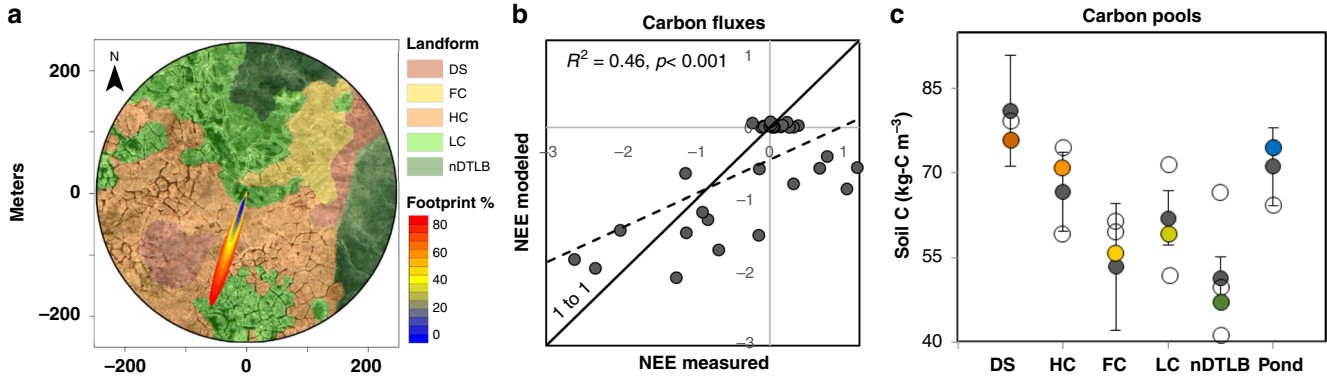

**Fig. 3 Validation of modeled carbon fluxes and carbon pools.** Monthly net ecosystem exchange (NEE) fluxes measured by the CARVE eddy covariance tower (71°19′22.72″N, 156°35′47.74″W, **a**) during 2011–2015, were compared with NEE fluxes simulated with DOS-TEM (dashed line in **b**). Negative NEE indicates carbon uptake, while positive NEE indicates loss. Footprint % indicates the accumulated percentage of measured NEE used to compare with modeled NEE, weighted by polygonal landform (DS drained slope, HC high center, FC flat center, LC low center, nDTLB nonpatterned drained thaw lake basins) using the Kormann and Meixner[93] flux footprint model (e.g., **a**). Modeled carbon pools (colored circles; **c**) were compared to 44 pedons collected (solid gray circles with standard error bars) and validated against 11 independent random subset of soil carbon pedons (open circles) measured on each respective landform across the Barrow Peninsula.

The rates of soil carbon accumulation varied by soil horizon (Supplementary Fig. 3). Although all landforms increased soil carbon at a relatively constant rate in the fibric horizon, the rate of accumulation differed between landforms in the humic horizon. Wet landforms continued to increase in carbon content, while moist and dry landforms either did not change (i.e., FC and HC) or lost carbon (i.e., DS). No notable changes in the mineral

horizon were identified (Supplementary Fig. 3), with the exception of FC that slowly increased at a rate of ~3.5 g C m$^{-2}$ year$^{-1}$.

We grouped landforms using a single cluster analysis, employing six key biogeophysical characteristics (i.e., water table depth, thaw depth, vegetation carbon, soil carbon, soil nitrogen, and percentage of clay) and re-parameterized and re-calibrated the model to evaluate the uncertainty associated with incrementally reducing the spatial resolution of the representation of tundra heterogeneity. The ensemble of "grouped" landform simulations (Fig. 4b) followed a similar change trajectory as individual landforms (Fig. 4a), identifying the magnitude of change among landforms to be greater than climate uncertainty (CCCMA A2 versus ECHAM5 B1). Similar to individual landform responses, wet landform groups "FC + LC + nDTLB" and "FC + LC + nDTLB + Pond" increased in soil carbon between 4523 and 3397 (7.7–5.8%) and between 3570 and 2570 (6.2–4.4%) g C m$^{-2}$, respectively (Fig. 4b). Landform groups moist "FC + LC" and dry "DS + HC" increased in soil carbon content between 3822 and 3126 (6.6–5.4%) and between 1356 and 907 (1.9–1.3%) g C m$^{-2}$, respectively (Fig. 4b). The "tundra-biome" landform group (parameterized with data from all landforms) well represented the mean trajectory across landforms, increasing between 2630 and 2094 (4.5–3.6%) g C m$^{-2}$ (Fig. 4b). Generally, the change in soil carbon by horizon for landform groups (Fig. 4b) was comparable to that identified by individual landforms (Fig. 4a and Supplementary Fig. 3).

**Impact of model spatial scale on soil carbon.** DOS-TEM was regionally extrapolated using all landform parameterizations across nine different scales (0.0009, 0.25, 0.5, 1, 2, 4, 8, 16, and 25 km$^2$), to evaluate the influence of model spatial scale on soil carbon dynamics. Input landform distributions were aggregated using a majority filter algorithm (i.e., resampling approach that reclassifies course resolution pixels using the most abundant underlying fine pixel values) from a 30-m resolution (i.e., 0.0009 km$^2$) tundra landform map[9]. Six non-overlapping 75 km$^2$ sub-regions of the Barrow Peninsula (Fig. 5b) were randomly selected to (i) standardize the area extent of the simulation domain and (ii) estimate model uncertainties associated with landform heterogeneity and model spatial scale, expressed as the bias and random error[57,58].

The coarsening of model spatial scales incrementally magnified uncertainties (Fig. 5a) and were directly linked to the misrepresentation of landform distribution (Fig. 6). The bias error became increasingly negative with scale, decreasing by −0.6% for every 1 km$^2$ coarsening of spatial scale. Bias errors ranged from

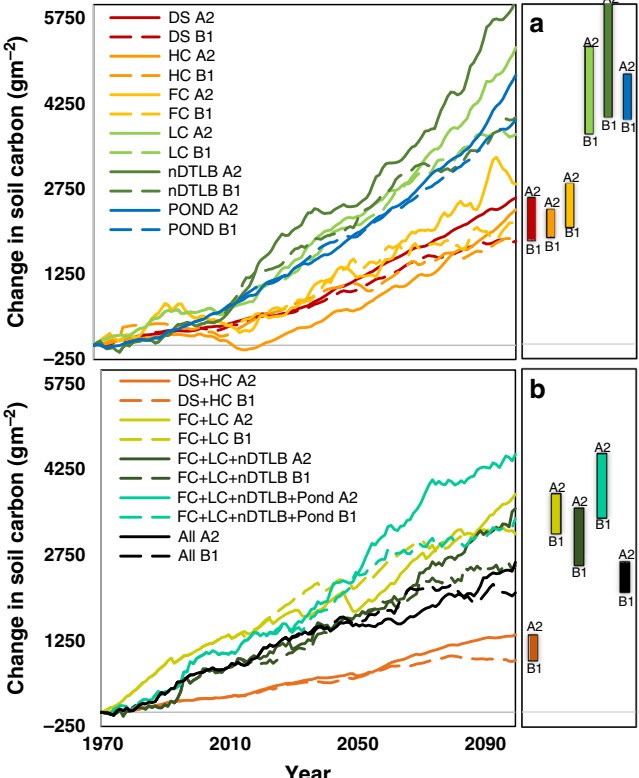

**Fig. 4 Projected change in soil carbon pools through 2100.** Simulated change in soil carbon among all polygonal tundra landforms (**a**; DS drained slope, HC high center, FC flat center, LC low center LC, nDTLB nonpatterned drained thaw lake basins) and landform groups (**b**) forced using CCCMA A2 and ECHAM5 B1 climate and emission pathways for our study region. Climate forcing for CCCMA A2 and ECHAM5 B1 are comparable to RCPs 6.0 and 4.5, respectively. Cluster analysis using key hydrologic and biogeochemical characteristics facilitated in the grouping of data for model reparameterization and re-calibration of landform groups (**b**).

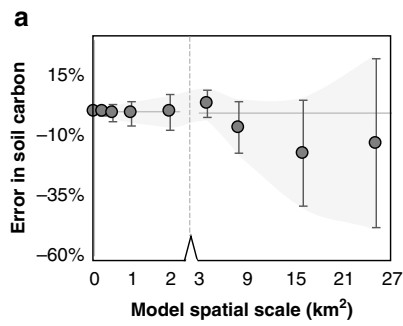

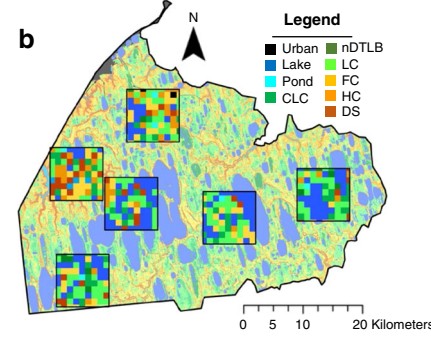

**Fig. 5 Error propagation with scale.** Errors of prediction in tundra soil carbon by 2100 in response to model spatial scale on the Barrow Peninsula (**a**). Simulations were computed using all tundra landforms and model spatial scales across six randomly located 75 km$^2$ subregions (**b**). Error metrics are visualized in **a**, as the deviation of simulations from 0 is representative of the bias error, and the range of simulation uncertainty (gray envelope) at each spatial scale represents the random error. Note axis break (dotted line) in **a**.

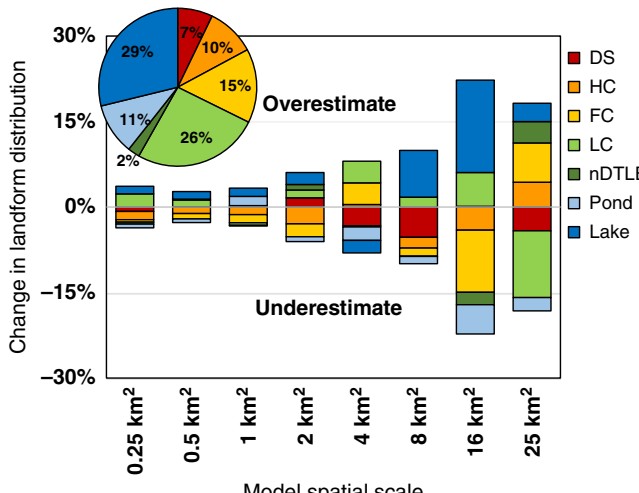

**Fig. 6 Misrepresentation of polygonal tundra landforms with scale.** Differences in landform distribution across spatial scales are relative to the highest resolution (0.0009 km²; pie chart). Data are representative of mean landform distributions across six subregions on the Barrow Peninsula. Negative and positive values indicate an overestimate and underestimate of polygonal tundra landforms, respectively.

0.5 to 11.9% associated with relatively fine (i.e., ≤4 km²) to coarse spatial scales (i.e., >4 km²), while random error became increasingly positive with scale, increasing by 1.4% for every 1 km² coarsening of spatial scale. Random errors ranged from 3.9 to 22.8% associated with fine to coarse-scale representation of tundra landforms.

Both the bias error and random error were significantly minimized at fine scales (Fig. 5a), as twenty-first century soil carbon was only overestimated by a maximum of 3.7 and ±7.4%, respectively. This is in contrast to coarser spatial scales as bias and random error sharply increased at 8, 16, and 25 km² by −6.1% (±10.7%), −17.0% (±22.1%), and −12.6% (±35.5%), respectively (Fig. 5a). The increase in spatial scale led to the overestimation in the area of low productivity thermokarst lakes (1.1% for every 1 km²) and underestimated wet productive landforms such as Ponds (−0.5% for every 1 km²) and LC polygons (−0.5% for every 1 km²; Fig. 6). This underestimation of wet landforms was particularly concerning as wet landforms have been regionally identified as those most sensitive to change[59–61], while representing a significant proportion of the regional carbon cycle[9,60,62,63].

**Influence of tundra heterogeneity and model spatial scale.** To evaluate the causes, consequences, and mitigation strategies for twenty-first century errors of prediction (i.e., bias and random error), we examined the combined influence of both tundra heterogeneity and model spatial scale. Correlation matrices clarified the potential causes of variable prediction errors, while hierarchical cluster analysis implemented using Euclidean distance and McQuitty linkage methods were used for grouping tundra heterogeneity and model spatial scales with similar errors of prediction to identify potential mitigation strategies or recommendations for future modeling applications.

Correlation matrices supported our presumption that an overestimation of lakes and underestimation of productive wet landforms altered the quantification of landscape-level soil carbon stocks, as bias error was strongly negatively correlated with lake cover (r = −0.98) and positively correlated with wet landforms (r = 0.94; Fig. 7). We found an inverse correlation in bias error as the prevalence of lake cover increased with spatial scale at the expense of nearly all other landforms, but in particular the

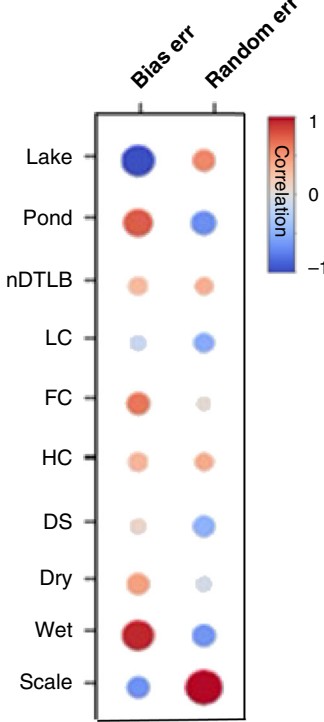

**Fig. 7 Pearson's correlations of uncertainty metrics (bias and random error) and spatial attributes.** The larger the bubble the greater the p value. Landform categories dry and wet include spatial data from "DS + HC" and "FC + LC + nDTLB+Pond", respectively. See Supplementary Fig. 3 for correlation bubble plots of all clusters.

landforms in low abundance such as tundra ponds (Figs. 6 and 7). Similar to the identified influence of spatial scale on random error (Fig. 4), correlations were highly positively related with model spatial scale (r = 0.99; Fig. 7), reinforcing the impact of coarsening model scale on uncertainty propagation.

Overall, bias error was linked with the misrepresentation of tundra landforms as spatial scale increased (Fig. 7 and Supplementary Fig. 4). Therefore, we next elucidated the influence of heterogeneity and scale on random error. Though random error was correlated with spatial scale, we explored the variability across tundra heterogeneity and scale. The lowest and highest random errors occurred at the finest (≤4 km²) and coarsest (≥16 km²) spatial scales, respectively (Fig. 8). Landform clusters include one or more landforms and landform groups needed to represent tundra heterogeneity on the Barrow Peninsula. Random error was constrained to ±4.5% by considering 5 or 6 tundra landform groups at fine scales. However, at coarse scales these heterogeneous groups also showcased the greatest errors (±28.9%) due to the high number of landforms parameterized within increasingly uncertain landform distributions as scale increased (Figs. 5 and 8). The lowest error among clusters was identified in landform cluster 2 (i.e., ±3.4%; dry and wet), likely due to biogeophysical similarities (i.e., soil anaerobicity, soil available nitrogen, productivity gradients) between dry versus wet landforms (Supplementary Table 1) and similar responses to climate change (e.g., Fig. 4a). Interestingly, even at coarse scales the error found in cluster 2 remained lower than all other landform clusters. Although the "tundra-biome" cluster 1 had a relatively low random error across spatial scales (Fig. 8), this result would not be directly transferable to other modeling applications as we leveraged (i) a robust dataset for model parameterization and (ii) high-resolution polygonal tundra landform maps, currently

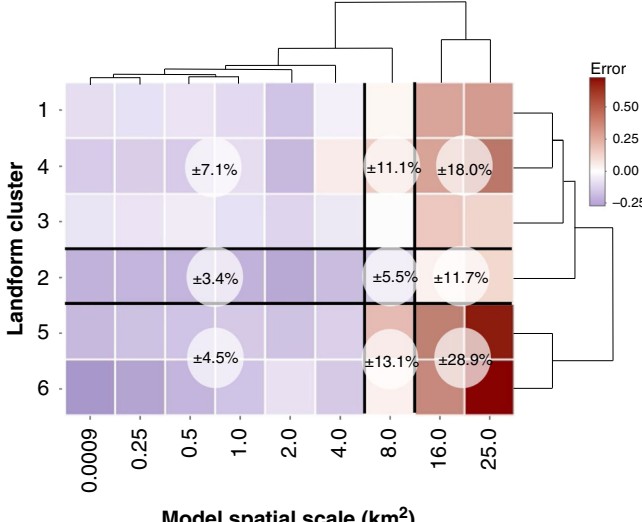

**Fig. 8 Heat-map of random error for all tundra heterogeneity and model spatial scales.** Warm to cool colors represent high to low random error (transformed to improve visualization). Hierarchical clustering grouped random error for all landform clusters (i.e., landforms and groups to represent the heterogeneity on the Barrow Peninsula) using a ~50% similarity cut-off for group membership. Mean random errors (transparent white circles) are presented for each landform cluster and model spatial scale.

unavailable across the Arctic for initializing and weighting model parameterization data. The importance of our data assimilation and landform weighting protocol was confirmed by testing the performance of a single unweighted landform parameterization (i.e., HC polygon) extrapolated across the Barrow Peninsula. We found random error to double (±15%) that of cluster 1 at fine scales (≤4 km$^2$) and nearly triple (±45%) at coarse scales (>8 km$^2$). Therefore, to best simulate dynamically changing carbon pools in permafrost soils, our analysis recommends a minimum of two landform groups (i.e., dry and wet) at a maximum model spatial scale of ≤4 km$^2$ (Fig. 8).

**Implications for modeling soil carbon dynamics in Arctic tundra.** Current uncertainties among Pan-Arctic model projections reflect inadequate spatial and temporal data needed to initialize, parameterize, and validate key Arctic ecosystem processes[55,56,64]. This study overcame many of these limitations by leveraging a legacy of data (1973–2016) collected from the data-rich Barrow Peninsula to constrain parameter, climate, and model uncertainties, to improve the representation of Arctic tundra heterogeneity across model spatial scales. We identify a scale-dependent balance between tundra heterogeneity and model spatial scale, linked with the decoupling of actual and simulated tundra landform distributions as spatial scales increased (Figs. 5 and 6). The scale-dependency of model process representation is supported by ground-based assessments, as the drivers of carbon dynamics vary across local (e.g., drainage conditions affecting aerobic/anaerobic processes), regional (e.g., vegetation distribution), and landscape scales (e.g., climate variability). Though we identified relatively minimal differences in carbon accumulation rates between polygonal tundra landforms, this was not necessarily surprising as Arctic coastal tundra landforms are relatively young (<5500 years)[24], often forming within drained lake basins underlain by the same initial soil substrates[65]. It is not until ice-wedge aggradation alters surface microtopography[14] that local changes in soil moisture, vegetation community composition, and

carbon and nitrogen fluxes incrementally alter soil carbon and nitrogen pools. Therefore, our results are ecologically consistent and computationally relevant, as the two recommended dry and wet tundra landforms are at opposite ends of the geomorphological succession spectrum[21] and found to not only be those most important to adequately represent tundra heterogeneity but also for minimizing prediction errors (bias and random error) while maximizing computational efficiency in Pan-Arctic modeling applications.

Because all models are subject to imprecision associated with imperfect observations[66], evaluating the influence of various sources of uncertainty in data-poor high-latitude ecosystems are of utmost importance[56,67,68]. The inherent randomness in natural systems over space and time was constrained by iteratively synthesizing nearly all possible data combinations to parameterize and simulate change in soil carbon across landforms and groups using relatively moderate climate scenarios for our study region (Supplementary Table 1 and Supplementary Fig. 2). Measurement error (i.e., imprecision of data) was overcome by capitalizing on the robust data collected across multiple projects, initiatives, and programs, thereby minimizing systematic error associated with sampling bias. Due to the difficulty quantifying the influences of cause-and-effect relationships among parameters, we constrained model structural uncertainty in this application by using a single process-based model, ensuring that ecosystem processes are constant among simulations

Despite constraining multiple sources of uncertainty, simulation limitations persist. For example, we did not explicitly simulate carbon dynamics in lakes on the Barrow Peninsula as regional carbon accumulation has been very low throughout the Holocene (~10 g C m$^{-2}$ year$^{-1}$)[69–71]. In contrast, lakes within yedoma deposits accumulate carbon at a rate of ~47 g C m$^{-2}$ year$^{-1}$[72]. This disparity highlights the importance of not only representing heterogeneous tundra landforms for minimizing uncertainties in Arctic carbon dynamics (Figs. 6 and 8) but also heterogeneous Arctic lakes. Although our results clearly identify the scale-dependency of simulated carbon dynamics, patterns of uncertainty propagation with scale are likely to persist across most spatially explicit model parameters (i.e., available nitrogen), specifically if models are unable to represent sub-grid-scale spatial variability. In addition, similar to many other models[73] DOS-TEM does not represent within-grid lateral flow (i.e., adjacent grid cells do not interact), making us unable to evaluate the influence of spatial scale on the lateral transport of carbon, nutrients, and water across the landscape. Recent catchment-scale modeling applications provide a more realistic representation of regional surface and subsurface tundra hydrology and terrestrial–atmosphere fluxes[74], with the potential to minimize scale-dependent uncertainties. Among the most conspicuous limitations in nearly all terrestrial and earth system models is in the inadequate representation of periglacial landscape evolutionary pathways (i.e., permafrost degradation) that will control spatial and temporal patterns of tundra wetting and drying[75]. Though DOS-TEM partially captures this process associated with the vertically resolved thawing of permafrost, these periglacial ecosystem dynamics are difficult to represent in gridded process models[76] because (i) data characterizing short- and long-term thermokarst (i.e., subsidence of the surface with permafrost thaw) dynamics are limited, (ii) permafrost degradation does not uniformly initiate across space and time[75,77], and (iii) degradation processes do not occur at a uniform rate for initiated thermokarst[77–79]. These small but prevalent thermokarst features may represent hot-spots of biogeomorphic change[80], yet not currently represented in any regional to Pan-Arctic process-based biogeochemistry model.

Due to these known limitations, this assessment focused on improving the representation of heterogeneous Arctic tundra

ecosystems in carbon cycle models. Similar to others[4,81,82], simulations indicate that elevated atmospheric $CO_2$ was the dominant driving factor for projected soil carbon sequestration (sum of carbon from litter, fibric, humic, and mineral soil horizons; Supplementary Fig. 5). This projected increase in soil carbon storage was in line with twenty-first century simulations in northern Alaska[83] and the Pan-Arctic[4]. As compared to Pan-Arctic Permafrost Carbon Network model intercomparison[4], TEM models consistently project higher rates of soil carbon sequestration than other models that do not include nitrogen dynamics, as net primary production is enhanced by nitrogen availability with soil warming. Compared to simulations from a previous version of TEM that used a single parameterization to represent Arctic tundra wetlands of northern Alaska[83], our soil carbon accumulation rates were still overestimated by as much as 75.4%. If these tundra wetland simulations[83] were implemented with at least two parameterizations (i.e., dry and wet), our findings estimate a threefold decrease in the error of prediction. Collectively, these results highlight that a slight increase in the representation of tundra heterogeneity in terrestrial and Earth System biogeophysical models may notably decrease uncertainties in projected frozen and seasonally thawed soil carbon dynamics in the Arctic.

## Discussion

High-latitude regions are among the largest sources of identified uncertainties to global climate projections[56,84]. Due to the spatial and temporal data limitations that propagate uncertainties in the Arctic, we leveraged the legacy of data measured near Utqiaġvik to evaluate the influence of local tundra heterogeneity on regional carbon modeling on the Barrow Peninsula. Comprehensive data assimilation and analysis suggested that bias and random errors will be significantly constrained by representing a minimum of two tundra landforms (dry and wet) at a maximum model spatial scale of ≤4 km². However, models capable of representing sub-grid processes, while accounting for landscape heterogeneity may overcome many of these errors of prediction. Models capable of accounting for landform distribution, shoulder season (i.e., zero-curtain freeze–thaw isothermal period) carbon dynamics[53], and thermokarst-driven landscape evolutionary dynamics may markedly minimize uncertainties in next-generation Arctic biogeochemistry models. Overall, our results indicate that model error will gradually decrease with the improved representation of tundra heterogeneity and spatial scale, and all efforts to advance the representation of permafrost-driven tundra heterogeneity in terrestrial and earth system models will significantly improve global climate change projections in response to thawing and decomposing permafrost carbon.

## Methods

**Dynamic organic soil in the terrestrial ecosystem model.** We use a process-based ecosystem model, the DOS-TEM, designed to simulate carbon and nitrogen pools and fluxes across water, vegetation, soils, and the atmosphere. DOS-TEM has been applied and validated at a range of scales and biomes[4,8,31,34,85], while accurately simulating the thickness and carbon content of organic-rich soils[86] in boreal[34] and Arctic landscapes[4,31,54,82]. The DOS-TEM is driven by climate, $CO_2$ forcing, vegetation-type distribution, soil texture, elevation, and disturbance. All climate input datasets were downscaled from 0.5° spatial resolution (CRU, www.cru.uea.ac.uk/) to 1 km resolution using the delta method[87] by the Scenarios Network for Alaska and Arctic Planning (www.snap.uaf.edu/). To represent the potential range of temporal variability between 2010 and 2100 simulations, two earth system models were used: (1) the CCCMA (version:3.1-t47, www.cccma.ec.gc.ca/data/cgcm3/) and (2) Max-Planck Institut fur Meteorologie (ECHAM5, version: echam-5.4.02, www.mpimet.mpg.de/en/wissenschaft/modelle/echam/). Each of these models conducted simulations to represent high (A2), moderate (A1B), and low (B1) $CO_2$ emission scenarios, respectively (IPCC, 2014). Downscaled historical and projected climate datasets include monthly mean air temperature, precipitation, vapor pressure, and surface incoming shortwave radiation from 1901 to 2010. Regional soil texture geospatial products used for model

initialization were derived from the median texture values by landform[9,13]. Elevation and wildfire disturbance were not important factors in this analysis as wet sedge graminoid tundra on the Barrow Peninsula is generally <8 m above sea level[9,88] and wildfires are extremely rare.

**Model initialization.** Each polygonal tundra landform was parameterized using vegetation and soil characteristics, including above and belowground vegetation nitrogen (N) and carbon (C), net primary productivity (NPP), gross primary productivity (GPP), N uptake, soil available N, $C_{Fall}$, $N_{Fall}$, organic horizon thickness, total fibric, humic, and mineral C (Supplementary Table 1). We used above and belowground biomass (g m$^{-2}$) estimates reported by Webber[21], to estimate vegetation C and N pools for each landform. Four regression analyses derived from datasets collected within ~1 km of sites sampled by Webber[21], over the summers of 2012–2014 (latitude: 71.28°, longitude: −156.60°) to estimate aboveground vegetation C and N. We developed the following regressions for vegetation C: $y = 0.458639x − 0.017729$ ($R^2 = 0.99$, $p < 0.0001$, $n = 278$), where the dependent and independent variables were vegetation C (g m$^{-2}$) and vegetation biomass (g m$^{-2}$), and vegetation N: $y = 0.025335x − 0.017729$ ($R^2 = 0.97$, $p < 0.0001$, $n = 278$), where the dependent and independent variables were vegetation N (g m$^{-2}$) and vegetation biomass (g m$^{-2}$). Belowground vegetation C and N were estimated using the following regressions: $y = 0.461639x − 0.223021$ ($R^2 = 0.99$, $p < 0.0001$, $n = 51$), where the dependent and independent variables were root C (g m$^{-2}$) and root biomass (g m$^{-2}$), and $y = 0.00859871x + 0.022262$ ($R^2 = 0.87$, $p < 0.0001$, $n = 51$), where the dependent and independent variables were root N (g m$^{-2}$) and root biomass (g m$^{-2}$; unburned graminoid tundra)[52]. Estimates of NPP (g m$^{-2}$ year$^{-1}$) and GPP (g m$^{-2}$ year$^{-1}$) specific to landform were calculated by Webber[21], while monthly $C_{Fall}$ or the amount of C dropped as litter was estimated by the following: $C_{Fall} = (NPP/vegetation carbon) \times (1/12)$. NPP and GPP parameterizations in Pond landform were weighted by the ratio of pond margin vegetation (i.e., reported by Webber[21]) to total pond area (~0.391) estimated from high-resolution vegetation map products[9,19] to improve the representativeness of vegetation productivity in ponds classified in tundra landform maps used to initialize DOS-TEM[9,13].

We synthesized 110 soil cores collected from this region to initialize soil carbon pools in all landforms across the Barrow Peninsula[22–24,30,38,39,49–51]. However, only 55 soil pedons (i.e., 100-cm soil cores) were used to estimate landform-specific soil carbon as the remaining were collected to a depth of <80 cm and/or did not have the necessary metadata for representing the geographic location or one of the three soil horizons (i.e., fibric, humic, mineral). High-resolution satellite imagery (i.e., World View 2 and Quickbird 2) was used to categorize each soil pedon with the associated landform (DS:5, HC:10, FC:4, LC:10, Mdw:20, and Pond:6). Soil horizon thickness is prescribed for fibric and amorphous soil layers, but DOS-TEM assumes 1 m of mineral soil thickness. Therefore, we estimate mineral soil carbon >100 cm using estimates[89] that approximate all soil carbon between 100 and 200 cm depth in this region at ~29 kg C m$^{-3}$. Estimates of total soil carbon (Supplementary Table 1) reflect updated values indicative of 1 m of mineral soil. Linear relationships were developed to estimate soil N from soil C using the following regression: $y = 0.049662x − 0.673615$ ($R^2 = 0.88$, $p < 0.0001$, $n = 134$), derived from data reported from Gersper[39], where the dependent and independent variables represent soil N (g N m$^{-2}$) and soil C (g C m$^{-2}$). Available soil N was computed as the sum of $NH_4$-N and $NO_3$-N within the rooting zone[90].

DOS-TEM was calibrated to ground-based data (i.e., benchmark values; Supplementary Table 1) using local climate, vegetation, and soil properties to estimate parameters that are difficult to measure in the field (i.e., decomposition rate-limiting parameters). Model calibration matched simulated ecosystem attributes to model benchmarks, which initiates by running DOS-TEM to an equilibrium state using the mean climate (i.e., air temperature, precipitation, vapor pressure, solar radiation) and $CO_2$ concentration from 1901 to 1930 for Utqiaġvik, Alaska. After tuning the model, equilibrium is achieved when model benchmarks match simulated values and little to no additional change in model behavior is observed during the simulation.

**Prediction error.** Error metrics were computed by assuming that the most accurate tundra landscape simulation corresponds with the highest representation of tundra landforms (i.e., all six landforms) and spatial distribution of landforms (i.e., 0.0009 km²), which we refer to as the *reference simulation*. Similar to others[91,92], the bias error was calculated as follows: $\frac{\sum_{i=1}^{n}(x-R)}{n}$, where $n$ is the total number of sub-regions, $x$ is a simulation at a given heterogeneity and scale (e.g., "landform cluster 4" at 1 km² resolution), and $R$ is the reference simulation, whereas random error was calculated as follows: $\sqrt{\frac{\sum_{i=1}^{n}(x-R)^2}{n}}$, by propagating variances across sub-regions, represented in the simplest form as the standard deviation.

**Model validation.** Gap-filled eddy covariance-derived NEE observations were acquired from the CARVE tower[48]. In situ flux data were reported in half-hour intervals between 2011 and 2015 and used to validate seasonal to inter-annual simulated patterns in carbon dynamics. Data were summarized to coincide with monthly time steps simulated by DOS-TEM. Measured NEE fluxes were linked to

landforms by computing the Kormann and Meixner[93] flux footprint model. Though we acknowledge the inherent uncertainty among eddy covariance tower footprint models and location biases, the Kormann and Meixner footprint model was selected due to (1) its ability to achieve a relatively simplistic two-dimensional analytical solution to the crosswind-distributed advection–diffusion equation[52] and (2) this footprint model has been widely applied in northern tundra ecosytems[94,95]. Thus our tower footprint is approximated using the height of the tower, wind speed, wind direction, friction velocity, Monin Ohbukov stability parameter, and the standard deviation of the cross-stream wind component. Similar to others[9,10], we identified polygonal tundra landforms using high-resolution 0.6 m pan-sharpened 2008 Quickbird 2 multispectral imagery and digitized polygonal land-forms in a 300-m radius of the tower. The proportion of landforms within 80% of the tower footprint were used to approximate NEE using DOS-TEM simulated fluxes and compare with eddy covariance-derived NEE observations (Fig. 3).

Of the 55 soil pedons that were able to represent carbon pools at depth, we randomly selected a subset of pedons for each landform (DS:1, HC:2, FC:2, LC:2, Mdw:3, and Pond:1) to independently validate modeled carbon pools (Fig. 3). The remaining 43 soil pedons were used to benchmark values for model calibration (Fig. 3 and Supplementary Table 1).

## Data availability

All data synthesized to parameterize DOS-TEM are summarized in Supplementary Table 1 and publically available via publications[21–24,30,38–40,49–51] or within data repositories[13,41–48,52]. Input or additional data subsets may be provided by M.J.L. upon request.

## Code availability

Custom program code for DOS-TEM may be provided by M.J.L. upon request. See provided correspondence details.

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

## Acknowledgements

This project was supported by (1) the Next-Generation Ecosystem Experiments (NGEE Arctic) project, funded by the Office of Biological and Environmental Research in the DOE Office of Science, and (2) the Alaska Climate Science Center through Grant/Cooperative Agreement G10AC00588 from the U.S. Geological Survey. We are grateful to the Ukpeaġvik Inupiat Corporation (UIC) for permitting/land access and the Barrow Arctic Science Consortium for logistical support. Any use of trade, firm, or product names is for descriptive purposes only and does not imply endorsement by the U.S. Government.

## Author contributions

M.J.L. designed the study, analyzed the data, integrated water layers in DOS-TEM, and wrote the manuscript. A.D.M. and E.S.E. refined the study design and discussion of ideas. H.G. implemented regional DOS-TEM applications. R.R. and S.Y. assisted with the integration of dynamic water layers in DOS-TEM. C.I., V.S., and S.D.W. collected the field data that helped to parameterize DOS-TEM. All authors reviewed the manuscript and made significant contributions to the writing.

## Competing interests

The authors declare no competing interests.
