## [Peer Review File · Nature Communications]

Reviewers' comments:

Reviewer #1 (Remarks to the Author):

Review Lara Paper

This paper aims to understand the impact of numerical model grid resolution on the future projection of soil carbon in the Arctic tundra, coarsening the grid resolution systematically by from 30 m to 5 km. They use DOS-TEM, which is a physically-based terrestrial ecosystem model with dynamics organic soil layers. There have been many studies that reported the importance of fine-scale heterogeneity associated with micro topography in polygonal tundra. They include fine-scale modeling studies in terms of thermal and hydrology (e.g., Liljedahl et al., 2016; Grant et al., 2017a) and carbon exchanges (e.g., Grant et al., 2017b). However, it has been uncertain how this fine-scale heterogeneity influences the large-scale carbon budget, or whether it matters. To my knowledge, this is the first paper that explored the impact of the grid resolution and the scale of heterogeneity. I think this is a significant contribution to the Arctic research community. I have several major concerns, and several suggestions.

- The finest-scale heterogeneity is 30 meter

Field studies has reported the critical impact of microtopography on carbon exchanges and soil biogeochemistry (e.g., Zona et al., 2011; Wainwright et al., 2015; Tas et al., 2018). Ice wedge polygons have microtopography that varies on the order of a few meters. For example, trough features - which are critical for water drainage - are a few meter width. LiDAR and satellite images are available at the resolution of several meters (Lara et al., 2015). Some modeling studies included the microtopography at sub-meter resolutions (Grant et al., 2017a and b). The authors need to justify why the 30 meter is sufficient to represent the fine-scale heterogeneity in ice-wedge polygons.

- The use of "uncertainty"

The authors define "uncertainty" as the difference between the courser-resolution results with the reference case (i.e., the finest-resolution simulation results). But they should have a better uncertainty metric (such as entropy). In Figure 5, for example, the "uncertainty" is negative, which does not make sense. Or the authors should use a precise term like "the difference from the finest resolution" or "the error in prediction". I don't think that the authors have good understanding of "uncertainty" in general. The authors should use formal terminology and precise terms (see for example in Scheidt et al., 2018).

- Organization

The manuscript should be better organized. For example, the authors highlight the potential loss of soil carbon in the introduction section, but their first projection result in Figure 4 shows the carbon gain. Then in the later section, the authors bring up CO₂ fertilization as a key factor for soil carbon increase. It is quite confusing. In addition, the authors discuss different sources of "uncertainty" later, which I think should be discussed in the introduction section. The authors need to better organize the texts to provide the clear flow of logic and to avoid confusions.

- Spatial aggregation strategy

To create the coarser-resolution grid from the original 30-m resolution map, the authors used a majority filter algorithm, which chooses the most common land class within each course-resolution grid. As the authors mentioned, the land class distribution changes at different resolutions, and some key land classes are overestimated/overestimated. It would be more meaningful if the authors could explore different spatial aggregation strategies. For example, in Figure 4a, there are two groups; wet and dry. It could be better to classify the landscape into just these two classes first at the 30-meter scale. Then upscaling would have less impact on the land class distribution than having more classes.

- The PCA results are confusing.

I don't see the significance of this PCA analysis, or it is confusing. The impact of particular land class at different resolution (e.g. lake cover) was discussed based on Figure 2B and Figure 3 (Line 180). In Figure 4 (mis-labeled; around Line 230), the heat map (or the error percentages) is pretty uniform. You could see the resolution at which the error increase in Figure 5a. The authors need a better explanation on why this analysis is meaningful.

Other comments.

L 38. The authors should have a correct description of ice-wedge polygon formation. Ice-wedge polygons were created by frost cracks, ice evolution within cracks, and associated soil geomorphic changes (Leffingwell et al., 2016)

L 40. Soil moisture should be spelled out since soil moisture is a key factor for thermal conductivity, and carbon cycling (Zona et al., 2011).

Figure 2b. The correlation is significant, but the modeled vs measured are not on the one-to-one line. Did this bias get corrected in the final simulations?

Figure 2c. Is the soil carbon vertically averaged? If so, what is the depth that is considered?

L125. Here the authors should explain why the soil carbon is increasing.

L 137. The models does not "decrease the heterogeneity"; it should be "incrementally reducing the resolution of representing the heterogeneity"

L157. It would be better to keep using "error" rather than "uncertainty" to be precise. The term "uncertainty" has to be carefully used. It does not make sense to have negative "uncertainty" in Figure 5.

L247. "Data uncertainty" vs "parameter uncertainty (i.e., imprecision of data)" are confusing. The authors need to use formal definitions in the uncertainty quantification literature.

References:

Grant, R. F., Mekonnen, Z. A., Riley, W. J., Wainwright, H. M., Graham, D., & Torn, M. S. (2017a). Mathematical modelling of arctic polygonal tundra with ecosys: 1. Microtopography determines how active layer depths respond to changes in temperature and precipitation. *Journal of Geophysical Research: Biogeosciences*, 122(12), 3161-3173.

Grant, R. F., Mekonnen, Z. A., Riley, W. J., Arora, B., & Torn, M. S. (2017b). Mathematical modelling of arctic polygonal tundra with ecosys: 2. Microtopography determines how CO₂ and CH₄ exchange responds to changes in temperature and precipitation. *Journal of Geophysical Research: Biogeosciences*, 122(12), 3174-3187.

Leffingwell, E. D. K. (1915), Ground-ice wedges: The dominant form of ground-ice on the north coast of Alaska, *J. Geol.*, 23, 635-654. L

Liljedahl, A. K., Boike, J., Daanen, R. P., Fedorov, A. N., Frost, G. V., Grosse, G., ... & Necsoiu, M. (2016). Pan-Arctic ice-wedge degradation in warming permafrost and its influence on tundra hydrology. *Nature Geoscience*, 9(4), 312.

Scheidt, C., Li, L., & Caers, J. (Eds.). (2018). *Quantifying uncertainty in subsurface systems* (Vol. 236). John Wiley & Sons.

Taş, N., Prestat, E., Wang, S., Wu, Y., Ulrich, C., Kneafsey, T., ... & Jansson, J. K. (2018). Landscape topography structures the soil microbiome in arctic polygonal tundra. *Nature communications*, 9(1),

777.

Zona, D., D. A. Lipson, R. C. Zulueta, S. F. Oberbauer, and W. C. Oechel (2011), Microtopographic controls on ecosystem functioning in the Arctic coastal plain, *J. Geophys. Res.*, 116, G00I08, doi:10.1029/2009JG001241.

Reviewer #2 (Remarks to the Author):

Review of the manuscript Lara et al.: Local-scale Arctic tundra heterogeneity affects regional-scale carbon dynamics

General comments

The authors leveraged soil carbon field observations between 1973 and 2016 collected on the Arctic Coastal Plain of Alaska and used the data to parameterize a Dynamic Organic Soil – Terrestrial Ecosystem Model (DOS-TEM). They used a set of polygonal tundra landforms and aggregation of these within a range of model spatial resolutions to assess the uncertainty caused by tundra heterogeneity and resolution in ecosystem models. Major claim of this manuscript is that the uncertainty of modeling soil carbon dynamics increases with coarsening of the chosen spatial scale while wet landforms are underestimated. The authors recommend a minimum of two landforms (dry and wet) and a maximum spatial scale of 4 square kilometers to reduce uncertainty in comparable modeling efforts

The study seems to be conducted carefully and the results are discussed well, although parts of their methods are missing, and the authors describe their data only in a general way, e.g. “key hydrological and biogeochemical characteristics”. Given the level of detail provided, it will be hard to reproduce the work.

My major concern is that the main conclusion, i.e. higher uncertainties with lower resolutions and landform representation, is not very surprising or novel. Generally, coarse resolutions are chosen due to computational and data constraints.

Technical comment: In parts of the manuscript, the authors use too long and complex sentences, too many brackets, partly repeat redundant information, and the numbering of figures is not matching.

Detailed comments

Abstract

#16

I cannot find the reference to the estimate of surface area in the main text.

#19

Stick to one unit when describing the scale range

#23

Reword “6-to-1 landform”

Introduction

#40-41

Avoid using “/”

#54

What do you mean by “data”? Please describe the observations used.

#57

Usage of “ ” is not necessary here.

#59

Please reword the question. Do you mean uncertainty of regional soil carbon dynamics in models (projections)? In general, the question already includes an answer, that is heterogeneity, in itself. Would it be interesting to ask if there is an over- or underestimation of certain landforms?

Study region

#71

Please provide a value for “meso-scale”.

#73-76

How do you distinguish between ponds and lakes?

It is tiring to attribute the estimated numbers to the long list of landforms, and it is unclear why you did not sort the list according to the surface area covered.

#80-82

Given the text, it is likely but not clear that this is sorted along a dry-wet gradient as you mention in the sentence before.

Results and Discussion

#90-91

Did you use soil cores with random diameter or pedons with standard 1 sqm at the surface, and hence 1 cubic meter?

#104

In my opinion “moderate” is a better word than “good” given coefficient of 0.46

#105

Wording “shoulder season” can be ambiguous, please use direct and clear words. Do you mean “fall”?

#114

“all landforms” can mean all possible definitions available were considered. It would be better to use “all defined landforms” or another more precise wording.

#116

Reference for trade-offs is missing.

#127

Indicate period (from to) for shown increases. Commonly you should use one decimal place for the shown percentages and “+” seem not necessary here.

#131

What do you mean by “lower than wet”?

#135-136

Still, in the figure it seems that there is a notable change in FC mineral carbon.

#137-140

Why do you only show 4 out of 6 key characteristics? Which are the 2 remaining?

#148-150

I do not understand the meaning of this sentence as there are differences between grouped and individual landforms in Suppl. Fig. 1

#161-164

I would recommend to shift the equations to the method section, separated from the text and add reference numbers as usual.

#178-179

I do not understand this sentence.

#238-244

To me this seems not very surprising or novel.

#267

The citation used for this assumption is referring to a study on European lakes. Is there also a study available from Arctic lakes?

Conclusions

#285-295

The main claim here is that Arctic studies suffer from lack of data and that the uncertainty in ecosystem modeling increases with spatial scale and decreases landform representation, which is not very surprising. It seems that you should try to add to the originality of the study. Did you try to elaborate more the cause and consequences of the underestimation of wet landforms, you showed?

Methods

A precise description of the observation data ("legacy of data collected between 1973 to 2016") is missing as well as multivariate statistics performed (clustering, PCA, correlations).

Confidence intervals are missing in the graphs that show changes in soil carbon.

Figures

The numbering of the figures is not continuous, partly doubled, and not matching with the text. I refer to lines for comments on the figures.

#47

Why did you choose a scale bar sub division of 35-560 km? Why not even numbers?

#52

Missing: Northing (view), scale, time, more precise indication of the location.

#94

Panel A: Indicate location in the caption. Explain footprint.

Panel B: Explain dashed line.

Caption: Indicate model used. For "colored circles" indicate n. Do you mean "solid grey circles" in Panel B?

#120

Can you show confidence intervals?

Caption: "through 2100" (which is also a single year) is not fitting to the figure (1970-2090). Please indicate the correct period here and elsewhere in figures.

#166

Panel A: The break in the x axis scale can be misleading, is there a better way to have a clear scaling?

Why did you use ... 9, 15, 21, 27?

Panel B: Add legend, scale and north arrow.

#230

Describe the clustering in the caption. The caption is not very clear to me. Can you reword and add more details, e.g. add the representation of landforms in clusters (6-1 is high to low?).

#262

See my comment #120

Supplement

#588

See my comment #120

Caption: I do not understand the last sentence.

#595

Information on the PCA is missing in the caption (explained variances).

#598

See my comment #120

Reviewer #3 (Remarks to the Author):

This manuscript assesses the potential errors in projected soil carbon change in the Arctic tundra due to scale errors, incomplete landform representation, and climate change uncertainty. Using DOS-TEM, the authors define and parameterize several key Arctic tundra landforms using data from a range of recent and historic datasets and then define a high-resolution (in scale and landform) simulation against which climate change simulations with larger spatial scales or a reduced number of landforms can be compared. Based on the model results, the authors identify a recommended minimum number of landforms (2, dry and wet) and spatial scale (< 4km²)

The results are interesting and are of relevance to the permafrost/Arctic terrestrial modeling community. The specific recommendation on the minimum number of landforms may be robust across models. The conclusion on scale may or may not be as transferable.

Overall, I appreciated the paper, but found it to be confusing in several places (some specific recommendations below). I didn't split up into major and minor points in my list below, though my main concern with the results is outlined in Issue 4. Presuming that the authors do not find any significant issues with the model simulations based on their assessment of my concerns in issue 4, I would find this paper suitable for publication after they address the minor revisions and a suggested additional edit of the entire manuscript for clarity.

Issues

1. Figure 2 caption. Footprint % not explained. What are the open circles in Figure 2C?

2. Line 106: Is it a minor underestimate? Hard to know how to qualify the underestimate. Probably best to remove the word minor.
3. Line 113: Should briefly define fribric and humic horizons for general readers that would not be familiar with soil classification terminology.
4. Line 117-118: Here or elsewhere, would be useful to provide at least some qualitative information about the climate change and CO₂ change for this modeled region for the different scenarios. With nothing more than low and high emissions, it is difficult to put the results in context. Related, I find it odd that for several landforms, there is very little difference in the change in soil carbon content across scenarios, despite the fact that there should be extreme differences in climate and CO₂ for these two scenarios. What is your explanation for that? It's strange in that it implies that for some landforms that 'any forcing' can generate the same net result. One possible explanation for a result like this is that the model was not actually in true equilibrium at the beginning of the run ... or that the manner in which the future forcing was applied provided an unrealistic kick to the system. Hopefully/presumably you have ruled out those two possibilities. Since the result of similar changes under strong differences in climate/CO₂ forcing is counterintuitive, I would suggest additional simulations with, for example, CO₂ held fixed and just climate forcing to try to better understand the response.
5. The model simulates an increase in soil carbon for all landforms, despite the strong warming and presumably pretty strong permafrost warming and thawing and a deepening of the active layer which one would think, at least for the dry soil cases, would lead to enhanced soil respiration and possibly soil carbon losses. The authors state that soil carbon accumulates due to CO₂ fertilization (higher litter inputs) of plants. As the authors note, other models, for example in the Permafrost Carbon Network model intercomparison project show a similar result. But, other models show a pretty different response. It would be useful to put the TEM results into context relative to the other PCN models. Is TEM on the high or low end of soil carbon change and vegetation carbon change in the PCN MIP?
6. Presumably, the area of each landform within a particular scale do not change during a simulation, though in principle in the real world they could evolve with climate forcing. Would be helpful to clarify that and discuss the consequences, already partly alluded to in the discussion section.
7. What is meant by model benchmarks? Usually obs data is the benchmark, but I think the data presented in Supp Table 1 is model average values or something like that.
8. Significant digits in soil carbon change results in paragraph including lines 125-135?
9. I'd suggest that the authors consider a different (or additional) method of displaying the data for Figure 4? As it is, it is pretty difficult to distinguish all the lines. What about showing a bar graph with just the change in carbon content by end of the century. This would be a cleaner picture. It seems to me that the details of the trajectory are not particularly relevant and are not really discussed in the text, so the end of the century result would be sufficient to make your points.
10. Figure numbers and ordering are all screwed up.
11. Significant digits in Table 1. For many fields, reporting out to way beyond reasonable accuracy. Would be a lot easier to read if sig digits were estimated and used.
12. Line 155: What is a 'majority filter algorithm'?
13. Figure 5 and Line 171: Is the term 'uncertainty' the right one here? It seems to me that the plot is showing the change in error (bias) induced by increasing the spatial scale. The paragraph in support of this paper would benefit from a rewrite for clarity. As written, I found it unnecessarily confusing and had to read several times to understand, partly due to unintuitive statements like - 'the bias error decreased by 0.64% for every 1km²'. From a mathematical perspective, this is correct. The bias error is becoming increasingly negative with scale. It would be more intuitive to state that the bias is increasing with scale, getting further from zero.
14. Lines 179-180: Again, this text is confusing. What exactly is meant by the statement that 'the coarsening of spatial scale overestimated low productivity thermokarst lakes'? Do the authors mean that the increase in scale lead to errors in the area of thermokarst lakes? Based on Figure 6, I do think that that is what the authors mean.
15. Line 192-194: I can't quite understand the PCA analysis. Perhaps expand explanation to be clearer about what was done. I read through this section several times and I can understand the points being made, but I have to be honest that I don't really understand how the authors got there. Apologies, but

again I think this points to my biggest criticism of the manuscript which is that it I found it hard to understand without multiple reads. This may be a consequence of the condensed nature of a Nature Communications paper.

16. Figure 7: After studying the figure for a while, I think I understand it. I'd suggest a bit more explanation in the figure caption. E.g., explain the percentages, explain the PCA score and relevance.

17. Line 291: It may be worth noting here that the recommendation on scale and landforms for accurate simulation is directly relevant to DOS-TEM. You could envision a model that is built with sub-grid tiles (one for each critical landform). If the area weights are accurate for each landform, then you could potentially run at much larger scales. In that type of model configuration, the degradation of accuracy would be due to the degraded (average) climate forcing at the larger scales and not due to inaccurate area totals of each landform.

More specifically, the methods, statistics, and organization of figures and tables need to be clarified. Reviewer #3 also suggested that the results of your modeling should be discussed in the context of other modeling runs.

Should further experimental data or analysis allow you to address these criticisms, we would be happy to look at a substantially revised manuscript. However, please bear in mind that we will be reluctant to approach the referees again in the absence of major revisions.

We are committed to providing a fair and constructive peer-review process. Do not hesitate to contact us if you wish to discuss the revision or if there are specific requests from the reviewers that you believe are technically impossible or unlikely to yield a meaningful outcome.

Reviewers' comments:

Very appreciative of all the reviewers/editors time that provided constructive comments, which we believe substantially improved the manuscript. We substantially revised the manuscript, specifically focusing on the simplification of unnecessarily complex sentence structure, reinforced the novelty of this study, altered our treatment of uncertainty terminology, and provided further analysis and discussion of the underlying drivers of projected carbon dynamics on the Barrow Peninsula.

Reviewer #1 (Remarks to the Author):

Review Lara Paper

This paper aims to understand the impact of numerical model grid resolution on the future projection of soil carbon in the Arctic tundra, coarsening the grid resolution systematically by from 30 m to 5 km. They use DOS-TEM, which is a physically-based terrestrial ecosystem model with dynamics organic soil layers. There have been many studies that reported the importance of fine-scale heterogeneity associated with micro topography in polygonal tundra. They include fine-scale modeling studies in terms of thermal and hydrology (e.g., Liljedahl et al., 2016; Grant et al., 2017a) and carbon exchanges (e.g., Grant et al., 2017b). However, it has been uncertain how this fine-scale heterogeneity influences the large-scale carbon budget, or whether it matters. To my knowledge, this is the first paper that explored the impact of the grid resolution and the scale of heterogeneity. I think this is a significant contribution to the Arctic research community. I have several major concerns, and several suggestions.

- The finest-scale heterogeneity is 30 meter

Field studies has reported the critical impact of microtopography on carbon exchanges and soil biogeochemistry (e.g., Zona et al., 2011; Wainwright et al., 2015; Tas et al., 2018). Ice wedge polygons have microtopography that varies on the order of a few meters. For example, trough features - which are critical for water drainage – are a few meter width. LiDAR and satellite images are available at the resolution of several meters (Lara et al., 2015). Some modeling studies included the microtopography at sub-meter resolutions (Grant et al., 2017a and b). The authors need to justify why the 30 meter is sufficient to represent the fine-scale heterogeneity in ice-wedge polygons.

You are correct. Research suggests microtopography and associated fine-scale polygonal landform heterogeneity influences above and belowground processes (motivation for this research; lines 36-44). Lara et al., 2015 provides good justification for this study as it explicitly evaluated the influence of carbon flux dynamics from polygonal landforms to landscape-scales. A key finding from Lara et al., 2015 was that carbon flux dynamics from microtopographic (cm resolution) heterogeneity was able to be closely approximated across spatial scales (plots, towers, and airborne flight lines) by the use of upscaled 30 m resolution polygonal landform distribution maps. Therefore modeling local to landscape-level ecosystem dynamics from our finest-scale (30 m spatial resolution) should be more than adequate in the Arctic Coastal Plain tundra.

However, it is important to note: 1) the greater spatial resolution the greater the parameterization data needed to represent the individual land cover features and types, which was or will not be possible in the foreseeable future, and 2) we will rapidly find ourselves entering the infamous coastline paradox.

- The use of “uncertainty”

The authors define “uncertainty” as the difference between the coarser-resolution results with the reference case (i.e., the finest-resolution simulation results). But they should have a better uncertainty metric (such as entropy). In Figure 5, for example, the “uncertainty” is negative, which does not make sense. Or the authors should use a precise term like “the difference from the finest resolution” or “the error in prediction”. I don’t think that the authors have good understanding of “uncertainty” in general. The authors should use formal terminology and precise terms (see for example in Scheidt et al., 2018).

Thank you for this clarification point. Agree that there are many specific definitions for “uncertainty”. We accepted your suggestion and changed the terminology regarding “uncertainty” throughout the manuscript to “error of prediction”.

- Organization

The manuscript should be better organized. For example, the authors highlight the potential loss of soil carbon in the introduction section, but their first projection result in Figure 4 shows the carbon gain. Then in the later section, the authors bring up CO₂ fertilization as a key factor for soil carbon increase. It is quite confusing. In addition, the authors discuss different sources of “uncertainty” later, which I think should be discussed in the introduction section. The authors need to better organize the texts to provide the clear flow of logic and to avoid confusions.

We have not only modified terminology, but focused on improving the clarification of key points throughout the manuscript to improve organization.

As you are likely well aware, there are many factors that influence soil carbon dynamics on short and long time scales. One of the key limitations, in Arctic tundra ecosystems is that we do not possess the necessary data or represent important Arctic ecosystem processes (yet to fully understand) to make confident predictions over the next 100 to 300 years. For example, there are many factors related to permafrost thaw/degradation and associated interactions with surface/subsurface hydrology that severely limit our confidence in future predictions in a rapidly warming world. I have updated the discussion to reflect these ideas and provide rationale for why we do not go into detailed interpretations of future carbon stocks (lines 279-282).

- Spatial aggregation strategy

To create the coarser-resolution grid from the original 30-m resolution map, the authors used a majority filter algorithm, which chooses the most common land class within each coarse-resolution grid. As the authors mentioned, the land class distribution changes at different resolutions, and some key land classes are overestimated/overestimated. It would be more meaningful if the authors could explore different spatial aggregation strategies. For example, in Figure 4a, there are two groups; wet and dry. It could be better to classify the landscape into just these two classes first at the 30-meter scale. Then upscaling would have less impact on the land class distribution than having more classes.

There are four primary methods to aggregate/resample geospatial data, 1) nearest neighbor, 2) cubic convolution, 3) bilinear interpolation, and 4) majority filtering. Each of these aggregation strategies have their strengths and weaknesses, and are best deployed under particular circumstances. Nearest neighbor is often used in classification aggregation due to its simplistic approach of using the cell center of the final raster output resolution and classifying it with the closest input cell (a method that is not interested in limiting information loss, developed to maximize computation efficiency). Cubic convolution is used in continuous raster data not typically in thematic maps such as classifications, it uses 16 neighboring cells to smooth the data into the final raster output product (not relevant or useful to our analysis). Bilinear interpolation is similar to cubic convolution but uses 4 neighboring cells to reclassify and smooth each pixel (another method to improve computational efficiency). Majority resampling is often used with thematic products as it uses the new resampled classification grid with distinct boundaries to determine the most popular class to be resampled in the output raster. This approach is the most widely used in thematic data and most intuitive for our particular scientific application, as it will determine the most appropriate landform for each spatial scale.

We now generally defined the majority filter algorithm on line 154.

Differences did exist between output maps produced by nearest neighbor versus majority filter, but these were very minimal (not worth drawing attention to this result). There is much literature describing the differences in geospatial aggregation strategies readers can refer to if interested.

- The PCA results are confusing.

I don't see the significance of this PCA analysis, or it is confusing. The impact of particular land class at different resolution (e.g. lake cover) was discussed based on Figure 2B and Figure 3 (Line 180). In Figure 4 (mis-labeled; around Line 230), the heat map (or the error percentages) is pretty uniform. You could see the resolution at which the error increase in Figure 5a. The authors need a better explanation on why this analysis is meaningful.

To eliminate confusion from the PCA analysis, we replaced this analysis with correlation matrices with a similar result.

We elected to keep the heat map as it clearly describes which landform clusters, groups, and scales correspond with the magnitude of simulation errors that could be expected as model scale and landform heterogeneity varies.

Other comments.

L 38. The authors should have a correct description of ice-wedge polygon formation. Ice-wedge polygons were created by frost cracks, ice evolution within cracks, and associated soil geomorphic changes (Leffingwel et al., 2016).

Thanks for the comment, Lachenbruch 1986 is one of the best papers to describe ice wedge morphology and dynamics. But we added another great reference that describes these processes (Sellmann et al., 1972).

L 40. Soil moisture should be spelled out since soil moisture is a key factor for thermal conductivity, and carbon cycling (Zona et al., 2011).

Agree soil moisture is a key component, we did not specifically add soil moisture in this list as it is a covariant of surface/subsurface hydrology. However we added a good Arctic soil moisture reference (Andresen et al., 2020) after “hydrology”.

Figure 2b. The correlation is significant, but the modeled vs measured are not on the one-to-one line. Did this bias get corrected in the final simulations?

We did not correct for the bias in our final simulations, as these data point to the lack of process representation of shoulder season thaw/freeze dynamics, as noted on line 103 (specifically during cold season carbon losses; NEE >0) .

Figure 2c. Is the soil carbon vertically averaged? If so, what is the depth that is considered?

The soil carbon data is vertically summed across all soil layers~100cm in depth. What is presented in this figure represents the average of all 100cm of soil carbon pedons available for each polygonal tundra landform. We provide details on lines 350 to 362.

L125. Here the authors should explain why the soil carbon is increasing.

We added a couple of sentences in the discussion highlighting our interpretation of why soil carbon is increasing (see lines 280-289).

L 137. The models does not “decrease the heterogeneity”; it should be “incrementally reducing the resolution of representing the heterogeneity”

The sentence was updated as suggested.

L157. It would be better to keep using “error” rather than “uncertainty” to be precise. The term “uncertainty” has to be carefully used. It does not make sense to have negative “uncertainty” in Figure 5.

We changed “uncertainty” to “error”, throughout.

L247. “Data uncertainty” vs “parameter uncertainty (i.e., imprecision of data)” are confusing. The authors need to use formal definitions in the uncertainty quantification literature.

We updated this section to follow terminology outlined by Uusitalo et al., 2015.

Uusitalo, L., Lehtikoinen, A., Helle, I. & Myrberg, K. An overview of methods to evaluate uncertainty of deterministic models in decision support. *Environmental Modelling and Software* (2015) doi:10.1016/j.envsoft.2014.09.017.

References:

Grant, R. F., Mekonnen, Z. A., Riley, W. J., Wainwright, H. M., Graham, D., & Torn, M. S. (2017a). Mathematical modelling of arctic polygonal tundra with ecosys: 1. Microtopography determines how active layer depths respond to changes in temperature and precipitation. *Journal of Geophysical Research: Biogeosciences*, 122(12), 3161-3173.

Grant, R. F., Mekonnen, Z. A., Riley, W. J., Arora, B., & Torn, M. S. (2017b). Mathematical modelling of arctic polygonal tundra with ecosys: 2. Microtopography determines how CO₂ and CH₄ exchange responds to changes in temperature and precipitation. *Journal of Geophysical Research: Biogeosciences*, 122(12), 3174-3187.

Leffingwell, E. D. K. (1915), Ground-ice wedges: The dominant form of ground-ice on the north coast of Alaska, *J. Geol.*, 23, 635–654. L

Liljedahl, A. K., Boike, J., Daanen, R. P., Fedorov, A. N., Frost, G. V., Grosse, G., ... & Necsoiu, M. (2016). Pan-Arctic ice-wedge degradation in warming permafrost and its influence on tundra hydrology. *Nature Geoscience*, 9(4), 312.

Scheidt, C., Li, L., & Caers, J. (Eds.). (2018). Quantifying uncertainty in subsurface systems (Vol. 236). John Wiley & Sons.

Taş, N., Prestat, E., Wang, S., Wu, Y., Ulrich, C., Kneafsey, T., ... & Jansson, J. K. (2018). Landscape topography structures the soil microbiome in arctic polygonal tundra. *Nature communications*, 9(1), 777.

Zona, D., D. A. Lipson, R. C. Zulueta, S. F. Oberbauer, and W. C. Oechel (2011), Microtopographic controls on ecosystem functioning in the Arctic coastal plain, *J. Geophys. Res.*, 116, G00I08, doi:10.1029/2009JG001241.

Reviewer #2 (Remarks to the Author):

Review of the manuscript Lara et al.: Local-scale Arctic tundra heterogeneity affects regional-scale carbon dynamics

General comments

The authors leveraged soil carbon field observations between 1973 and 2016 collected on the Arctic Coastal Plain of Alaska and used the data to parameterize a Dynamic Organic Soil – Terrestrial Ecosystem Model (DOS-TEM). They used a set of polygonal tundra landforms and aggregation of these within a range of model spatial resolutions to assess the uncertainty caused by tundra heterogeneity and resolution in ecosystem models. Major claim of this manuscript is that the uncertainty of modeling soil carbon dynamics increases with coarsening of the chosen spatial scale while wet landforms are underestimated. The authors recommend a minimum of two landforms (dry and wet) and a maximum spatial scale of 4 square kilometers to reduce uncertainty in comparable modeling efforts

The study seems to be conducted carefully and the results are discussed well, although parts of their methods are missing, and the authors describe their data only in a general way, e.g. “key hydrological and biogeochemical characteristics”. Given the level of detail provided, it will be hard to reproduce the work.

Thanks for your comments, we clarified many sections throughout the manuscript to improve clarity and reproducibility.

My major concern is that the main conclusion, i.e. higher uncertainties with lower resolutions and landform representation, is not very surprising or novel. Generally, coarse resolutions are chosen due to computational and data constraints.

We would agree that coarse resolutions are selected in land or earth system models due to computational constraints. However the specific influence of spatial scale and tundra heterogeneity is unknown. This information would be particularly important for selecting appropriate criteria to best represent vulnerable permafrost ecosystems for minimizing uncertainties and maximizing computational efficiency in circumpolar-scale model applications (see lines 237-246).

Technical comment: In parts of the manuscript, the authors use too long and complex sentences, too many brackets, partly repeat redundant information, and the numbering of figures is not matching.

We simplified the text throughout and corrected issues with figure numbering.

Detailed comments

Abstract

#16

I cannot find the reference to the estimate of surface area in the main text.

These references are added on line 37 after “(Fig. 2)”.

#19

Stick to one unit when describing the scale range

It is tricky business sticking to one unit as models are applied at a range of scales and are not exactly comparable. Many pan-arctic models are applied using lat/long grid cells,

which are vary in area dependent on distance from the equator. We did not change these units, but added text on line 64 clarifying this point.

#23

Reword "6-to-1 landform"

We rephrased this sentence and deleted this detail from the abstract.

Introduction

#40-41

Avoid using "/"

Deleted the use of "/" throughout.

#54

What do you mean by "data"? Please describe the observations used.

replaced "data" with, "measurements collected (described in Model Initialization)"

#57

Usage of " " is not necessary here.

Deleted

#59

Please reword the question. Do you mean uncertainty of regional soil carbon dynamics in models (projections)? In general, the question already includes an answer, that is heterogeneity, in itself. Would it be interesting to ask if there is an over- or underestimation of certain landforms?

Thank you for your suggestions. We added discussion of over- or underestimation of landforms in discussion, and rephrased this section with a hypotheses. Because the introduction reinforced the idea that heterogeneity is important, it was redundant to restate this as a question. The text was modified as follows, "Due to the strong control of polygonal landforms on ecosystem structure and function, we hypothesize the local-scale representation of tundra heterogeneity (*defined here as the total number of landforms represented in the model*) will markedly influence regional-scale soil carbon projections. However, the potential impact of tundra heterogeneity and model spatial scale (i.e. grid size) on arctic tundra carbon dynamics remains highly uncertain, yet paramount for reducing model uncertainties spanning the Pan-Arctic."

Study region

#71

Please provide a value for "meso-scale".

Provided a definition and modified the sentence to, "This continuous permafrost region is characterized by meso-scale (tens to hundreds of square kilometers) drained thaw lake

basins (DTLBs) and interstitial tundra^{9,38}, which are composed of a mosaic of fine-scale polygonal tundra landforms (tens to hundreds of square meters)."

#73-76

How do you distinguish between ponds and lakes?

References provided on line 74, outline the specific details in which landform characterization was performed by Lara et al., 2018a and b. Large lakes ($\leq 100,000 > 90$ ha), medium lakes ($\leq 90 > 20$ ha), small lakes ($\leq 20 > 1$ ha), and ponds (≤ 1 ha). All lake classes were lumped in this study.

It is tiring to attribute the estimated numbers to the long list of landforms, and it is unclear why you did not sort the list according to the surface area covered.

Updated as suggested.

#80-82

Given the text, it is likely but not clear that this is sorted along a dry-wet gradient as you mention in the sentence before.

Updated this section to improve clarity.

Results and Discussion

#90-91

Did you use soil cores with random diameter or pedons with standard 1 sqm at the surface, and hence 1 cubic meter?

We scraped together all relevant data, which were collected with different methods and core diameters. However, all data were presented along vertical depth profiles along 5 to 10cm increments ≥ 80 cm. The fibric and humic organic layers were typically < 30 cm in depth and the remaining ~ 70 cm was mineral soil. We used Bockheim and Hinkel (2007), to estimate the remaining carbon content of all our pedons, as they found all mineral soils in this region can be approximated at ~ 29 kgC m⁻³. Much of this detail is outlined on lines 356-368.

Bockheim, J. G. & Hinkel, K. M. The importance of 'deep' organic carbon in permafrost-affected soils of arctic Alaska. *Soil Sci. Soc. Am. J.* (2007) doi:10.2136/sssaj2007.0070N.

#104

In my opinion "moderate" is a better word than "good" given coefficient of 0.46

We changed this sentence, which now only refers to "good correspondence" for observations outside of the "shoulder season".

#105

Wording “shoulder season” can be ambiguous, please use direct and clear words. Do you mean “fall”?

Updated this sentence to, “ Although we identified good correspondence with modeled and measured NEE for most of our observations, DOS-TEM underestimated respiratory losses during the zero-curtain seasonal freeze and thaw isothermal period (e.g. September and October)⁵⁴, resulting in a underestimate of the 1 to 1 line ($R^2= 0.46$, $p< 0.001$, Fig. 3A,B).”

#114

“all landforms” can mean all possible definitions available were considered. It would be better to use “all defined landforms” or another more precise wording.

updated as suggested.

#116

Reference for trade-offs is missing.

This sentence was simplified/deleted to avoid confusion.

#127

Indicate period (from to) for shown increases. Commonly you should use one decimal place for the shown percentages and “+” seem not necessary here.

updated as suggested.

#131

What do you mean by “lower than wet”?

Elaborated this sentence, which now reads, “Ponds gained soil carbon at a slightly lower rate than wet landforms...”

#135-136

Still, in the figure it seems that there is a notable change in FC mineral carbon.

updated line 133, which now reads, “No notable changes in the mineral horizon were identified (Supplementary Fig. 2), with the exception of FC which slowly increased ~ 3.5 gC m⁻² year⁻¹.”

#137-140

Why do you only show 4 out of 6 key characteristics? Which are the 2 remaining?

Added all six.

#148-150

I do not understand the meaning of this sentence as there are differences between grouped and individual landforms in Suppl. Fig. 1

updated this sentence, which now reads, “Generally, the change in soil carbon by horizon (Fig. 4B) was comparable to that identified by individual landforms (Fig. 4; Supplementary Fig. 2).”

#161-164

I would recommend to shift the equations to the method section, separated from the text and add reference numbers as usual.

Updated as suggested.

#178-179

I do not understand this sentence.

Rephrased and linked with the topic sentence. “The coarsening of model spatial scales incrementally magnified uncertainties (Fig. 5A), and were directly linked to the misrepresentation of landform distribution (Fig. 6).”

#238-244

To me this seems not very surprising or novel.

Rephrased this generic summary statement to highlight the novelties of this study. Sentence now reads, “However results provide recommendations for minimizing prediction errors (bias- and random error) and maximizing computation efficiency and the representation of Arctic tundra heterogeneity in Pan-Arctic modeling applications.”

#267

The citation used for this assumption is referring to a study on European lakes. Is there also a study available from Arctic lakes?

We included two additional studies from the North American Arctic.

Whalen, S. C. & Cornwall, J. C. Nitrogen, phosphorus, and organic carbon cycling in an arctic lake. *Can. J. Fish. Aquat. Sci.* 42, 797–808 (1985)

Billings, W. D. Carbon balance of Alaskan tundra and taiga ecosystems: Past, present, and future. *Quat. Sci. Rev.* 6, 165–177 (1987).

Conclusions

#285-295

The main claim here is that Arctic studies suffer from lack of data and that the uncertainty in ecosystem modeling increases with spatial scale and decreases landform representation, which is not very surprising. It seems that you should try to add to the originality of the study. Did you try to elaborate more the cause and consequences of the underestimation of wet landforms, you

showed?

We updated the conclusion and discussion to emphasize the novelties of our findings, while putting results into context with Land Carbon project and Permafrost Carbon Network modeling efforts.

Methods

A precise description of the observation data (“legacy of data collected between 1973 to 2016”) is missing as well as multivariate statistics performed (clustering, PCA, correlations).

removed PCA analysis from the study to improve clarity.

Confidence intervals are missing in the graphs that show changes in soil carbon.

Because our model is deterministic not probabilistic we are unable to generate confidence intervals.

Figures

The numbering of the figures is not continuous, partly doubled, and not matching with the text. I refer to lines for comments on the figures.

Thanks, not sure what happened to the numbering scheme. Corrected.

#47

Why did you choose a scale bar sub division of 35-560 km? Why not even numbers?

updated as suggested.

#52

Missing: Northing (view), scale, time, more precise indication of the location.

Changed caption to, “Oblique aerial photograph of the dominant polygonal tundra landforms on the Arctic Coastal Plain of Alaska. Photograph acquired in August 2008, from southeast (~135° azimuth) of 71°16'46.02"N, 156°25'45.35"W.”

#94

Panel A: Indicate location in the caption. Explain footprint.

Added location and footprint% definition.

Panel B: Explain dashed line.

Caption: Indicate model used. For “colored circles” indicate n. Do you mean “solid grey circles” in Panel B?

Updated as suggested.

#120

Can you show confidence intervals?

Caption: "through 2100" (which is also a single year) is not fitting to the figure (1970-2090).

Please indicate the correct period here and elsewhere in figures.

Similar to the above response... because our model is deterministic not probabilistic, we are unable to generate confidence intervals. However we added variability in simulations associated with climate/emission scenarios for each landform simulation.

Simulations are through 2100. Without a minimum of 10 vertical grid lines, even spacing of integers across 130 years is not possible. Using the 4 intervals 1970, 2010, 2050, 2090 appears to be sufficient.

#166

Panel A: The break in the x axis scale can be misleading, is there a better way to have a clear scaling? Why did you use ... 9, 15, 21, 27?

Panel B: Add legend, scale and north arrow.

We used continuous numerical scales before and after the axis break. Without the axis break, all the values prior to 2km are overlapping and difficult to visualize. We ended the continuous scale at 27 to be able to view the error bar at 25km. We added a dotted line onto the axis break, to draw attention to this axis discontinuity; described in the caption.

Added landform legend, scale bar, and north arrow.

#230

Describe the clustering in the caption. The caption is not very clear to me. Can you reword and add more details, e.g. add the representation of landforms in clusters (6-1 is high to low?).

Updated the figure caption as suggested.

#262

See my comment #120

Deterministic not probabilistic, unable to generate "confidence intervals".

Supplement

#588

See my comment #120

Deterministic not probabilistic, unable to generate "confidence intervals".

Caption: I do not understand the last sentence.

deleted as it was unnecessary information

#595

Information on the PCA is missing in the caption (explained variances).

Deleted for clarity.

#598

See my comment #120

Deterministic not probabilistic, unable to generate “confidence intervals”.

Reviewer #3 (Remarks to the Author):

This manuscript assesses the potential errors in projected soil carbon change in the Arctic tundra due to scale errors, incomplete landform representation, and climate change uncertainty. Using DOS-TEM, the authors define and parameterize several key Arctic tundra landforms using data from a range of recent and historic datasets and then define a high-resolution (in scale and landform) simulation against which climate change simulations with larger spatial scales or a reduced number of landforms can be compared. Based on the model results, the authors identify a recommended minimum number of landforms (2, dry and wet) and spatial scale (< 4km²)

The results are interesting and are of relevance to the permafrost/Arctic terrestrial modeling community. The specific recommendation on the minimum number of landforms may be robust across models. The conclusion on scale may or may not be as transferable.

Overall, I appreciated the paper, but found it to be confusing in several places (some specific recommendations below). I didn't split up into major and minor points in my list below, though my main concern with the results is outlined in Issue 4. Presuming that the authors do not find any significant issues with the model simulations based on their assessment of my concerns in issue 4, I would find this paper suitable for publication after they address the minor revisions and a suggested additional edit of the entire manuscript for clarity.

Thank you for your comments, suggestions, and critiques, we 1) clarified details regarding the relevancy of our results to other modeling applications, 2) simplified the text throughout, 3) removed the PCA analysis and replaced with a more transparent correlation matrix, and 4) added additional regional simulation analysis using detrended atmospheric CO₂ data.

Issues

1. Figure 2 caption. Footprint % not explained. What are the open circles in Figure 2C?

We updated the figure caption to clarify Footprint% and open circles.

2. Line 106: Is it a minor underestimate? Hard to know how to qualify the underestimate. Probably best to remove the word minor.

Removed “minor”

3. Line 113: Should briefly define fibric and humic horizons for general readers that would not be familiar with soil classification terminology.

Appreciative of this comment, we added a general description, “These soil horizons are vertically stratified and vary in the degree of organic matter decomposition.”

4. Line 117-118: Here or elsewhere, would be useful to provide at least some qualitative information about the climate change and CO₂ change for this modeled region for the different scenarios. With nothing more than low and high emissions, it is difficult to put the results in context.

Added the following text, “These scenarios projected an increase in air temperature (6.96 to 5.72 °C), precipitation (182 to 215 mm), and atmospheric CO₂ (509 to 214 ppm) between 1970 and 2100.” In addition, we included a new corresponding supplemental figure 1.

Related, I find it odd that for several landforms, there is very little difference in the change in soil carbon content across scenarios, despite the fact that there should be extreme differences in climate and CO₂ for these two scenarios. What is your explanation for that?

Climate scenarios CCCMA A2 and ECHAM5 B1 represent the highest and lowest scenarios for this region, but I would not classify the differences “extreme”. The most profound differences occurs with atmospheric CO₂ concentration (see Supplemental Figure 1).

In response to your comment, the following text was added on lines 237-246, “Though we identified relatively minimal differences in carbon accumulation rates between polygonal tundra landforms, this was not necessarily surprising as Arctic coastal tundra landforms are relatively young (<5,500 years)²⁴, often forming within drained lake basins underlain by the same initial soil substrates⁶⁶. It is not until ice-wedge aggradation alters surface microtopography¹⁴ that local changes in soil moisture, vegetation community composition, and carbon and nitrogen fluxes incrementally alter soil carbon and nitrogen pools. Therefore our results are ecologically consistent and computationally relevant, as the two recommended *dry and wet* tundra landforms are at opposite ends of the geomorphological succession spectrum²¹ and found to not only be those most important to adequately represent tundra heterogeneity, but for minimizing prediction errors (bias- and random error) while maximizing computational efficiency in Pan-Arctic modeling applications.

It's strange in that it implies that for some landforms that ‘any forcing’ can generate the same net result. One possible explanation for a result like this is that the model was not actually in true equilibrium at the beginning of the run ... or that the manner in which the future forcing was applied provided an unrealistic kick to the system. Hopefully/presumably you have ruled out those two possibilities.

This similarity can also imply the limitations in our ability to account for finer-scale ecosystem processes such as seasonal patterns of snow depth distribution and redistribution that may influence soil thermal properties differently among landforms and across the landscape. A process that DOS-TEM was unable to represent.

We ensured our model was indeed in equilibrium prior to simulations.

Since the result of similar changes under strong differences in climate/CO₂ forcing is counterintuitive, I would suggest additional simulations with, for example, CO₂ held fixed and just climate forcing to try to better understand the response.

Thanks for this suggestion as it improved the interpretability of our regional simulations. We added detrended atmospheric CO₂ to regional simulations. It should now be clear to the reader that CO₂ enrichment had a significant role in carbon accumulation through 2100 (Supplemental Fig. 4).

5. The model simulates an increase in soil carbon for all landforms, despite the strong warming and presumably pretty strong permafrost warming and thawing and a deepening of the active layer which one would think, at least for the dry soil cases, would lead to enhanced soil respiration and possibly soil carbon losses. The authors state that soil carbon accumulates due to CO₂ fertilization (higher litter inputs) of plants. As the authors note, other models, for example in the Permafrost Carbon Network model intercomparison project show a similar result. But, other models show a pretty different response. It would be useful to put the TEM results into context relative to the other PCN models. Is TEM on the high or low end of soil carbon change and vegetation carbon change in the PCN MIP?

In addition to details provided above, we added the following to the discussion, “This projected increase in soil carbon storage was in line with 21st century simulations in northern Alaska⁸³ and the Pan-Arctic⁴. As compared to Pan-Arctic Permafrost Carbon Network model intercomparison⁴, TEM models consistently project higher rates of soil carbon sequestration than other models that do not include nitrogen dynamics, as net primary production is enhanced by nitrogen availability with soil warming.”

6. Presumably, the area of each landform within a particular scale do not change during a simulation, though in principle in the real world they could evolve with climate forcing. Would be helpful to clarify that and discuss the consequences, already partly alluded to in the discussion section.

This logic is correct. We added a bit of text describing the rationale of focusing on errors versus predicting 21st century carbon stocks... citing this and other reasons as limitations we were unable to consider (line 272).

7. What is meant by model benchmarks? Usually obs data is the benchmark, but I think the data presented in Supp Table 1 is model average values or something like that.

The figure caption now reads, “Observations used to benchmark DOS-TEM simulations from all available data sources. We recalculated benchmarks for landform groups by weighting data by the prevalence of each landform. GPP=Gross Primary Productivity....”

8. Significant digits in soil carbon change results in paragraph including lines 125-135?

We updated significant digits throughout.

9. I'd suggest that the authors consider a different (or additional) method of displaying the data for Figure 4? As it is, it is pretty difficult to distinguish all the lines. What about showing a bar graph with just the change in carbon content by end of the century. This would be a cleaner picture. It seems to me that the details of the trajectory are not particularly relevant and are not really discussed in the text, so the end of the century result would be sufficient to make your points.

We updated this figure to include bars indicating the range of soil carbon change associated with CCCMA A2 and ECHAM5 B1 by the end of the century.

10. Figure numbers and ordering are all screwed up.

Not sure what happened to the numbering scheme. Corrected.

11. Significant digits in Table 1. For many fields, reporting out to way beyond reasonable accuracy. Would be a lot easier to read if sig digits were estimated and used.

We deleted table 1 and replaced with a correlation matrix to improve clarity without the use of PCA analysis.

12. Line 155: What is a 'majority filter algorithm'?

following this terminology, we added, "(i.e resampling approach that reclassifies coarse resolution pixels using the most abundant underlying fine pixel values)."

13. Figure 5 and Line 171: Is the term 'uncertainty' the right one here? It seems to me that the plot is showing the change in error (bias) induced by increasing the spatial scale. The paragraph in support of this paper would benefit from a rewrite for clarity. As written, I found it unnecessarily confusing and had to read several times to understand, partly due to unintuitive statements like – 'the bias error decreased by 0.64% for every 1km²'. From a mathematical perspective, this is correct. The bias error is becoming increasingly negative with scale. It would be more intuitive to state that the bias is increasing with scale, getting further from zero.

Appreciative of your comments. We 1) changed the use of "uncertainty" throughout the manuscript to errors of prediction to avoid potential mischaracterization or confusion of our results, 2) rephrased much of the text throughout to improve clarity as suggested.

14. Lines 179-180: Again, this text is confusing. What exactly is meant by the statement that 'the coarsening of spatial scale overestimated low productivity thermokarst lakes'? Do the authors mean that the increase in scale lead to errors in the area of thermokarst lakes? Based on Figure 6, I do think that that is what the authors mean.

This section has been updated for clarity. In particular, this sentence was rephrased, "The increase in spatial scale lead to the overestimation in the area of low productivity thermokarst lakes (1.1% for every 1 km²) and underestimated wet productive landforms such as Ponds (-0.5% for every 1 km²) and low-center polygons (-0.5% for every 1 km²; Fig. 6)."

15. Line 192-194: I can't quite understand the PCA analysis. Perhaps expand explanation to be clearer about what was done. I read through this section several times and I can understand the

points being made, but I have to be honest that I don't really understand how the authors got there. Apologies, but again I think this points to my biggest criticism of the manuscript which is that it I found it hard to understand without multiple reads. This may be a consequence of the condensed nature of a Nature Communications paper.

To improve clarity, we pivoted and removed the PCA analysis from our paper and replaced with correlation matrices which served the same purpose (describe patterns among simulation errors with spatial data) but and improved interpretability.

16. Figure 7: After studying the figure for a while, I think I understand it. I'd suggest a bit more explanation in the figure caption. E.g., explain the percentages, explain the PCA score and relevance.

Updated the figure and elaborated on the figure caption, "*Figure 8: Heat-map of random error for all tundra heterogeneity and model spatial scales. Warm to cool colors represent high to low random error (transformed to improve visualization). Hierarchical clustering grouped random error for all landform clusters (include one or more landforms and groups to represent the tundra landscape) using a ~50% similarity cut off for group membership. Mean random errors (transparent white circles) are presented for each landform cluster and model spatial scale.*"

17. Line 291: It may be worth noting here that the recommendation on scale and landforms for accurate simulation is directly relevant to DOS-TEM. You could envision a model that is built with sub-grid tiles (one for each critical landform). If the area weights are accurate for each landform, then you could potentially run at much larger scales. In that type of model configuration, the degradation of accuracy would be due to the degraded (average) climate forcing at the larger scales and not due to inaccurate area totals of each landform.

You are correct, we added a couple of sentences in the discussion and the conclusion highlighting this point. "Comprehensive data assimilation and analysis suggested bias and random errors will be significantly constrained by representing a minimum of two tundra landforms (dry and wet) at a maximum model spatial scale of ≤ 4 km². However, models capable of representing sub-grid processes, while accounting for landscape heterogeneity may overcome many of these errors of prediction."

REVIEWER COMMENTS

Reviewer #2 (Remarks to the Author):

Second review of the manuscript Lara et al.: Local-scale Arctic tundra heterogeneity affects regional-scale carbon dynamics

The authors provided a careful point-to-point revision of their manuscript which greatly improved the robustness and readability of the study. I found only one (minor) issue that was not solved from my previous comments. All other problems that I mentioned were corrected or clarified. Therefore I can recommend publication of the manuscript after minor revision.

#16

My previous comment was: "I cannot find the reference to the estimate of surface area in the main text". In your revision you added references after Fig 2, but not related to your statement (65 % composed of polygons) that I only can find in the abstract. In my understanding an abstract should not state anything that is not explicitly mentioned in the text.

Kind regards
Boris Biskaborn

Reviewer #3 (Remarks to the Author):

Overall, the authors did a good job of addressing the reviewer comments and I believe that all of my original comments have been adequately addressed. I have just a few additional comments (minor revisions) here.

Line 63: Suggest rewording for clarity - We evaluate the error of prediction in 21st century Arctic soil carbon stocks associated with spatial scale by running parallel DOS-TEM simulations with a range of resolved landforms (1 to 6) and a broad range of spatial scales (30m to 50km).

It's surprising that the difference in climate change between A2 and B1 is so small (6.96C vs 5.72C). Intuitively, I would have expected a bigger difference, but I guess that's what it is. RCP8.5 (roughly equivalent to A2) vs RCP4.5 (roughly equivalent to B1), which were the scenarios used in the Permafrost Carbon Network, showed a pan-Arctic temperature changes of ~8C vs 3.5C, more of the large range that I would have expected. I guess it would be difficult at this stage to rerun with RCPs instead of SRES scenarios. SRES scenarios are really very old at this point. I wouldn't mind so much except that the climate change differences in the particular simulations used don't appear to really reflect the consensus difference in climate change between high and low emissions scenarios. So, what to do about this? If it is easy, then I'd run with RCPs or even the newer SSPs. But, I realize that that could be a relatively large amount of work for little gain. Another alternative is to simply drop one of the scenarios and avoid the topic of putting this into the context of climate change uncertainty. It's not a big focus of the paper anyway. Perhaps the message could be restated to avoid much discussion of the relative impact of climate change uncertainty. Another option would be put these actual climate change scenarios into a bigger context. That is, you could calculate the climate change for this region from a much bigger set of CMIP5 and/or CMIP6 models and see where these two scenarios sit relative to other scenarios.

Can you clarify what is meant by this statement: "Compared to regional single model parameterization of Arctic tundra wetlands of northern Alaska, our soil carbon accumulation rates were overestimated by as much as 75.4%. Nevertheless, if tundra wetland simulations were implemented with at least two

parameterizations (i.e. dry and wet), we estimate a 3-fold decrease in this prediction error. " I had a hard time understanding what the message was in these sentences. Maybe just reword to make it clearer what configurations of the model you are referring to when you note the bias. And, perhaps make it clearer what exactly you are comparing to (another model? Why should that model result be considered a reference result?

REVIEWER COMMENTS

Overall, we thank the reviewers for their thoughtful comments that substantially improved the manuscript.

Reviewer #2 (Remarks to the Author):

Second review of the manuscript Lara et al.: Local-scale Arctic tundra heterogeneity affects regional-scale carbon dynamics

The authors provided a careful point-to-point revision of their manuscript which greatly improved the robustness and readability of the study. I found only one (minor) issue that was not solved from my previous comments. All other problems that I mentioned were corrected or clarified. Therefore I can recommend publication of the manuscript after minor revision.

#16

My previous comment was: "I cannot find the reference to the estimate of surface area in the main text". In your revision you added references after Fig 2, but not related to your statement (65 % composed of polygons) that I only can find in the abstract. In my understanding an abstract should not state anything that is not explicitly mentioned in the text.

Thanks for your thoughtful comments Boris, we altered line 36 to now read, "The tundra on the Arctic Coastal Plain of Alaska is highly heterogeneous (Fig. 1), nearly 65% of this landscape is composed of an intricate network of ice-wedge polygon landforms (Fig. 2)^{12,13} developed by ground ice aggregation and degradation associated with frost heaving and ground subsidence¹⁴.

Reviewer #3 (Remarks to the Author):

Overall, the authors did a good job of addressing the reviewer comments and I believe that all of my original comments have been adequately addressed. I have just a few additional comments (minor revisions) here.

Line 63: Suggest rewording for clarity - We evaluate the error of prediction in 21st century Arctic soil carbon stocks associated with spatial scale by running parallel DOS-TEM simulations with a range of resolved landforms (1 to 6) and a broad range of spatial scales (30m to 50km).

This sentence (line 63) was rephrased to, "We evaluate the error of prediction in 21st century Arctic soil carbon stocks associated with tundra heterogeneity and spatial scale by running parallel DOS-TEM simulations with a range of resolved tundra landforms (6 to 1) and spatial scales (30 m to 25 km²)."

It's surprising that the difference in climate change between A2 and B1 is so small (6.96C vs 5.72C). Intuitively, I would have expected a bigger difference, but I guess that's what it is. RCP8.5 (roughly equivalent to A2) vs RCP4.5 (roughly equivalent to B1), which were the scenarios used in the Permafrost

Carbon Network, showed a pan-Arctic temperature changes of ~8C vs 3.5C, more of the large range that I would have expected.

After a careful comparison of regionally specific RCP and our SRES scenarios, your presumption that the temperature change should have been greater was correct. Our climate simulations CCCMA A2 and ECHAM5 B1 were actually most similar to RCPs 6.0 and 4.5 for the Barrow Peninsula. RCP 8.5 projects a much warmer scenario. We outline the various steps we took to rectify this issue below:

I guess it would be difficult at this stage to rerun with RCPs instead of SRES scenarios. SRES scenarios are really very old at this point. I wouldn't mind so much except that the climate change differences in the particular simulations used don't appear to really reflect the consensus difference in climate change between high and low emissions scenarios. So, what to do about this? If it is easy, then I'd run with RCPs or even the newer SSPs. But, I realize that that could be a relatively large amount of work for little gain. Another alternative is to simply drop one of the scenarios and avoid the topic of putting this into the context of climate change uncertainty. It's not a big focus of the paper anyway. Perhaps the message could be restated to avoid much discussion of the relative impact of climate change uncertainty. Another option would be put these actual climate change scenarios into a bigger context. That is, you could calculate the climate change for this region from a much bigger set of CMIP5 and/or CMIP6 models and see where these two scenarios sit relative to other scenarios.

Although, I would be interested in simulated outputs from RCP 8.5 and 4.5, it would not be feasible to use them in our application as we ran our simulations with downscaled 1km outputs. Such a downscaling procedure may be indeed a large undertaking, but more importantly, the climate focus would be a bit beyond the scope of this application.

We have now added a new supplemental figure, which compares the projected decadal mean temperature and precipitation among the five best-performing CMIP5 climate models (using the range of emission scenarios RCP 8.5 and 4.5) for Utqiagvik, AK. Including this figure, improves the perspective of our climate change projections within the context of other more commonly used scenarios. In addition, to maintain clarity throughout the manuscript we updated our terminology for defining climate scenarios from extreme and conservative to simply high and low (lines 114, 128, 656), and explicitly mention multiple times in the text that our climate projections are comparable to RCP 6.0 and 4.5 (lines 116, 123, 657).

Can you clarify what is meant by this statement: "Compared to regional single model parameterization of Arctic tundra wetlands of northern Alaska, our soil carbon accumulation rates were overestimated by as much as 75.4%. Nevertheless, if tundra wetland simulations were implemented with at least two parameterizations (i.e. dry and wet), we estimate a 3-fold decrease in this prediction error. " I had a hard time understanding what the message was in these sentences. Maybe just reword to make it clearer what configurations of the model you are referring to when you note the bias. And, perhaps make it clearer what exactly you are comparing to (another model? Why should that model result be considered a reference result?

Appreciate your well-founded points of contention. We compared our simulations to a prior version of DOS-TEM simulations as our justification for using these outputs as a reference result (included in our updated response).

This sentence was updated to, “Compared to simulations from a previous version of TEM that used a single parameterization to represent Arctic tundra wetlands of northern Alaska⁸⁴, our soil carbon accumulation rates were still overestimated by as much as 75.4%. If these tundra wetland simulations⁸⁴ were implemented with at least two parameterizations (i.e. dry and wet), our findings estimate a 3-fold decrease in the error of prediction.”

REVIEWERS' COMMENTS:

Reviewer #3 (Remarks to the Author):

The authors have satisfactorily addressed my comments.

David Lawrence